# Damages during February, 6-24 2017 Çanakkale earthquake swarm

Ramazan Livaoğlu[1], Mehmet Ömer Timurağaoğlu[1], Cavit Serhatoğlu[1], Mahmud Sami Döven[2]

[1]Department of Civil Engineering, Engineering Faculty, Uludağ University, Bursa, Turkey
[2]Department of Civil Engineering, Engineering Faculty, Dumlupinar University, Kütahya, Turkey

*Correspondence to*: Ramazan Livaoğlu (rliva@uludag.edu.tr)

**Abstract.** On February 6, 2017 earthquake swarm began at the western end of the Turkey. This has been the first recorded swarm at Çanakkale region since continuous seismic monitoring began in 1970. The number of located earthquakes increased during the next ten days. This paper describes the output of a survey carried out in the earthquake prone towns of Ayvacık, Çanakkale, Turkey, in February 2017 after the earthquakes. Observations were made on site regarding traditional buildings at

the rural area of Ayvacık. A description of the main structural features and their effects on the most frequently viewed damage modes were made according to in plane, out of plane behaviour of the wall regarding construction practice, connection type etc. It was found that there were no convenient connection details like cavity-ties or sufficient mortar strength resulting in decreased and/or lack of lateral load bearing capacity of the wall. Furthermore, distribution maps of damaged/undamaged buildings according to villages, damage ratios, structures and damage levels are generated. Distribution maps showed that

damage ratio of structures is higher in villages close to epicenter and decrease away from epicenter except Gülpınar where cultural accumulation and development level affect the construction quality.

## 1 Introduction

On February 6, 2017, a swarm of earthquakes began at the western end of Turkey at 06:51 local time. This has been the first recorded swarm in this region of Turkey since continuous seismic monitoring began in 1970. The number of located

earthquakes increased during the next ten days and earthquakes bigger than Mw=5.0 were experienced five times (Table 1). These data were taken from DEMP (Disaster and Emergency Management Presidency) report. The largest peaks from these medium-sized earthquakes occurred twice (Mw =5.3) at different local times on February, 6 2017 at a depth of 7 and 9.83 km, respectively (Table 1). The earthquakes and aftershocks that took place in this area between February 6 and 24, 2017 are shown in Fig. 1a. A total of 1930 earthquakes (M>2.0) occurred until February 24. The distribution of the epicentres of the events

and their magnitudes proved the earthquake swarm characteristics. Fig. 2 shows the evidence of the swarm. This graph depicts distribution of both Magnitude vs. Occurrence date and Magnitude vs. Cumulative number of earthquakes via time between February 6 and 16.

According to the active fault map prepared by MRE (General Directorate of Mineral Research and Exploration), these earthquakes occurred as strike-slip normal fault in the region near the Tuzla segment of Kestanbol fault and Gülpınar fault

(Fig. 1b). There were five villages which were closer than 5 km to the epicentre of the earthquakes. Around 30 villages were

struck by the earthquakes, which damaged nearly 2600 structures, and fortunately there were no casualties. The closest county centre, where there is almost no critical damage and loss of life, is approximately 15~20 km far from the epicentres of the earthquakes.

5 Table 1. Parameters of Ayvacık Earthquakes (DEMP, 2017)

| Date | Local time | Latitude | Longitude | Depth (km) | Magnitude ($M_{L,w}$) | Max Acc.-PGA (g) |
|---|---|---|---|---|---|---|
| 06.02.2017 | 06:51 | 39.5495 | 26.1370 | 14.12 | 5.3 | 0.078 (N-S) |
| 06.02.2017 | 13:58 | 39.5303 | 26.1351 | 8.70 | 5.3 | 0.103 (N-S) |
| 07.02.2017 | 05:24 | 39.5205 | 26.1570 | 6.24 | 5.2 | 0.090 (E-W) |
| 10.02.2017 | 11:55 | 39.5236 | 26.1946 | 7.01 | 5.0 | 0.038 (N-S) |
| 12.02.2017 | 16:48 | 39.5336 | 26.1700 | 7.00 | 5.3 | 0.089 (E-W) |

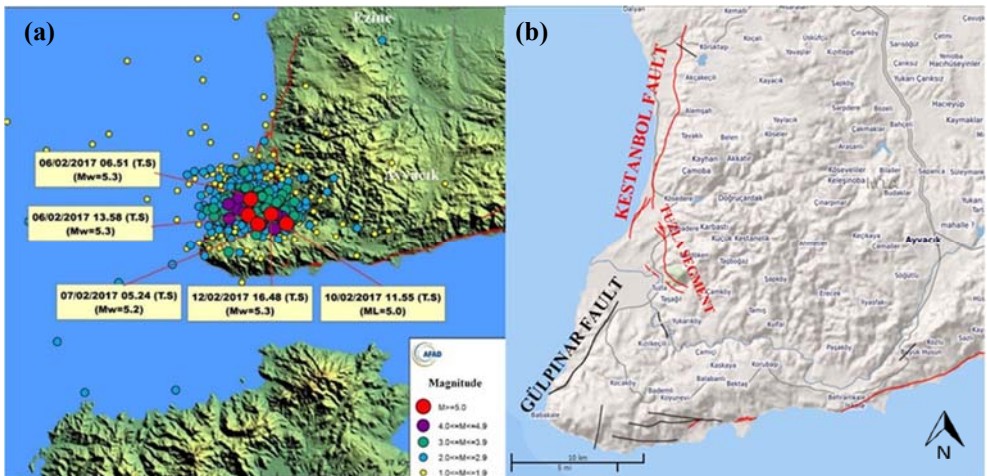

**Figure 1: (a) February 6-24 2017 Çanakkale –Ayvacık Earthquakes and aftershocks (DEMP, 2017) (b) Active fault map for Ayvacık, Çanakkale (Emre et al., 2013)**

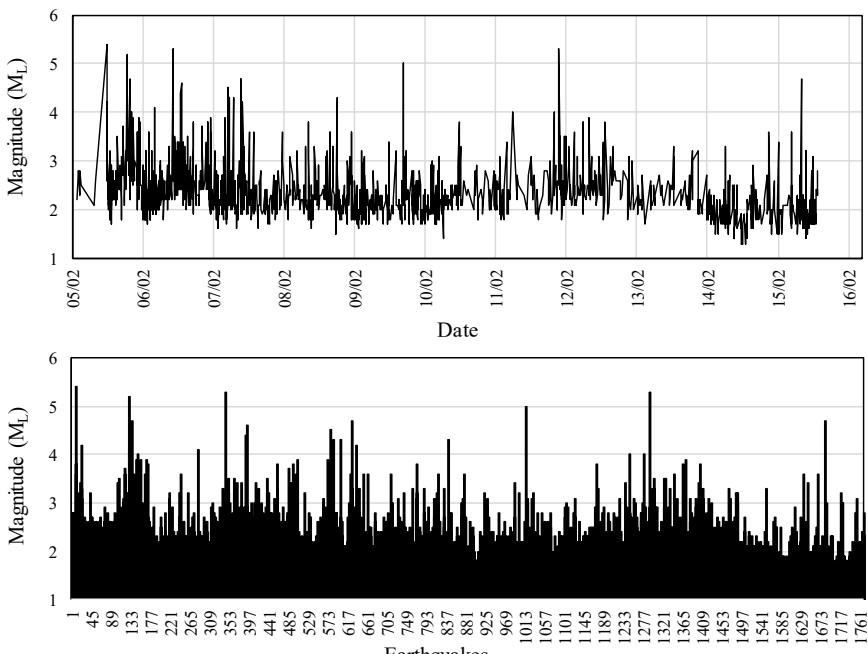

**Figure 2: Distribution of the 6-16 February Gülpınar/Ayvacık Earthquakes via date and cumulative number up to related date**

In Turkey, there are many different construction styles for supporting systems. More than 90% of these are reinforced concrete in city centres. However, traditional rural domestic style supporting systems are very distinctive; resulting from cultural attributes related to the availability of material and climate conditions of the building site. Timber is also one of the main materials preferred in building framed mansions and dwellings, especially in the Black Sea region of Turkey and in other hill/mountain side regions where timber is abundant. In any case, stone is a material that can be easily found, and lack of timber leaves people no choice but to use more stone in construction details. However, stone is not a convenient material in earthquake prone areas, because of its unit weight and being difficult to process. Timber also has an extensive history as a main structural "Hatıl" reinforcing element in rubble stone, brick and adobe houses, the predominant types of houses for ordinary people especially in rural areas (Hughes, 2000) (Fig. 3).

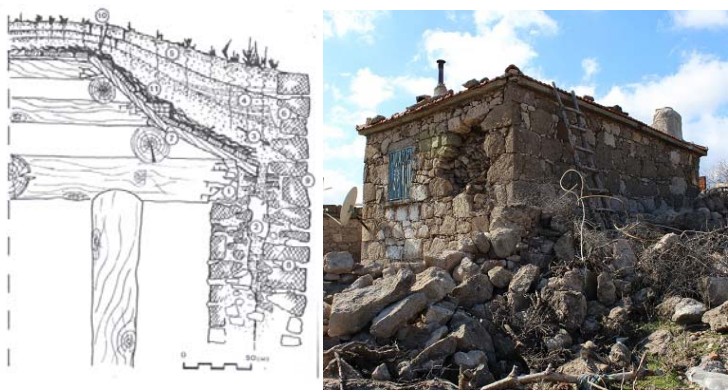

**Figure 3: Typical wall construction detail (Hughes, 2000) and typical view of a dwelling from the region**

In the reconnaissance area, observations showed that the construction materials and skills were extremely deficient. Modern materials and techniques were only used in a small portion of the observed region. Moreover, cement mortar between stones was not used in almost 50% of the walls. There were a few buildings in which reinforced concrete elements were partly or fully used. Curing of concrete is still not practiced as an integral part of the concreting process. The concrete blocks are of poor quality because of poor quality of the concrete, lack of compaction and very little or no curing. The existing building types in the area are shown in Fig. 4.

A field reconnaissance was carried out by four authors immediately after the earthquakes on February 12-17, for a period of five days and the observations were reported in the present paper. The authors also experienced the Mw=5.3 earthquake on 12 February during their observations. The objective of field reconnaissance was to record the causes of the damage patterns observed in the buildings, mainly in the rural areas affected by the earthquake swarm. The paper discusses the seismological aspects of the earthquakes, describes the classifications of buildings in the area and elaborates on the performance of various building types during the earthquakes and evaluates the damage distributions according to villages, damage ratios, structures and damage levels.

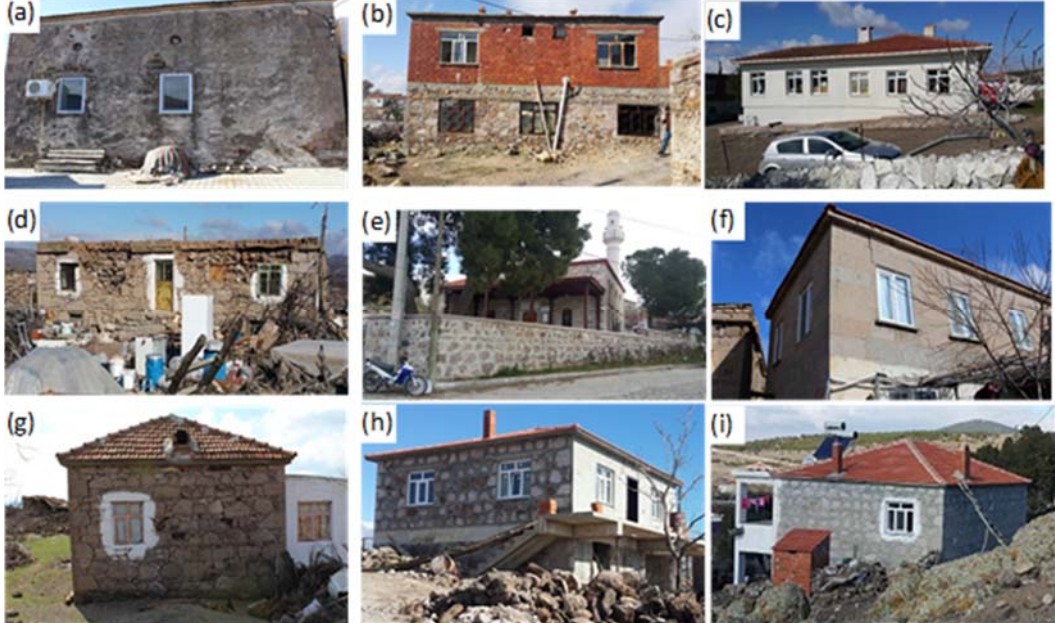

**Figure 4: Existing building types in reconnaissance area: a) hatıl dwelling b) stone and brick in cement mortar, c) engineered RC building d) hatıl building with heavy roof, e) historical masonry with cut stone, f) cut stone without mortar, g) stone without mortar, h), i) stone in cement mortar with reinforced concrete.**

## 2 Seismicity of the Region

Turkey is an earthquake-prone country which is located on seismically active regions in the 'Alpine–Himalayan Earthquake belt', and its complex deformation is a result of the continental collision between African and Eurasian plates (Fig. 5a). The major neotectonic elements of the region are the dextral North Anatolian Fault Zone (NAFZ), the Sinistral East Anatolian Fault Zone (EAFZ) and the Aegean–Cyprus Arc, which forms a convergent plate boundary between the Afro-Arabian and Anatolian plates (Gürer and Bayrak, 2017). The geological events in the region such as plate motions, seismic activities and crustal deformations are attributed to these major neotectonic entities (Bozkurt, 2001).

In this study, the region of north-west Anatolia has been investigated from both land and sea. This region is one of the most important active seismic and deformation regions between Eurasian and African tectonic plates. The region is affected by both the strike-slip tectonic regime, which is a general characteristic of NAFZ, and the extensional regime of west Anatolian block. The most effective earthquake within the instrumental period (after 1900) around the region are the Aegean Sea earthquake (M=7.2) that occurred in 1981, Ayvacık-Çanakkale earthquake (M=7.0) in 1919, and Edremit gulf earthquake (M=6.8) in 1944 (KOERI, 2017) as shown in Figure 5b.

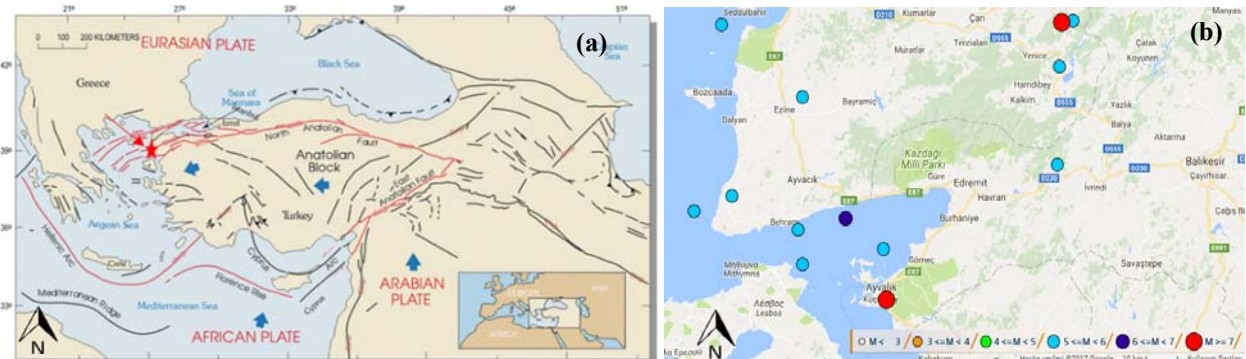

**Figure 5: (a) Simplified Tectonic Map of Turkey (USGS, 2005) (b) The most effective earthquake within the instrumental period around Ayvacık region**

**3 Ground motions and Response spectra**

An instrument situated in a low-rise appurtenant building adjacent to the local office of the Forestry Operation Directorate of Çanakkale Ayvacık recorded the shock as being 15~25 km away from the hypocenters. The acceleration components of three earthquakes recorded by this instrument are given in Fig. 6. As seen in this figure, the peak ground accelerations (amax) are 70~110 mG (cm/s$^2$) in the North-South (N-S) direction, 70~90 mG in the East–West (E-W) direction, and 20~30 mG in the vertical (U-D) direction for the shocks bigger than $M_w$=5. According to earthquake zoning map of Turkey prepared by General Directorate of Disaster Affairs in 1996, the seismic zone of the city of Çanakkale is classified as 1, where the probability of exceeding an effective peak ground acceleration of 0.4g is 10 percent in 50 years or the return period is 475 years (TEC 2007). As can be seen in Fig. 6, the peak value of acceleration was maximal in the N–S component and occurred as 110 cm/s$^2$. It should be noted that peak ground acceleration didn't exceed the seismic hazard defined to be 0.4g for this area in the seismic zone map of Turkey.

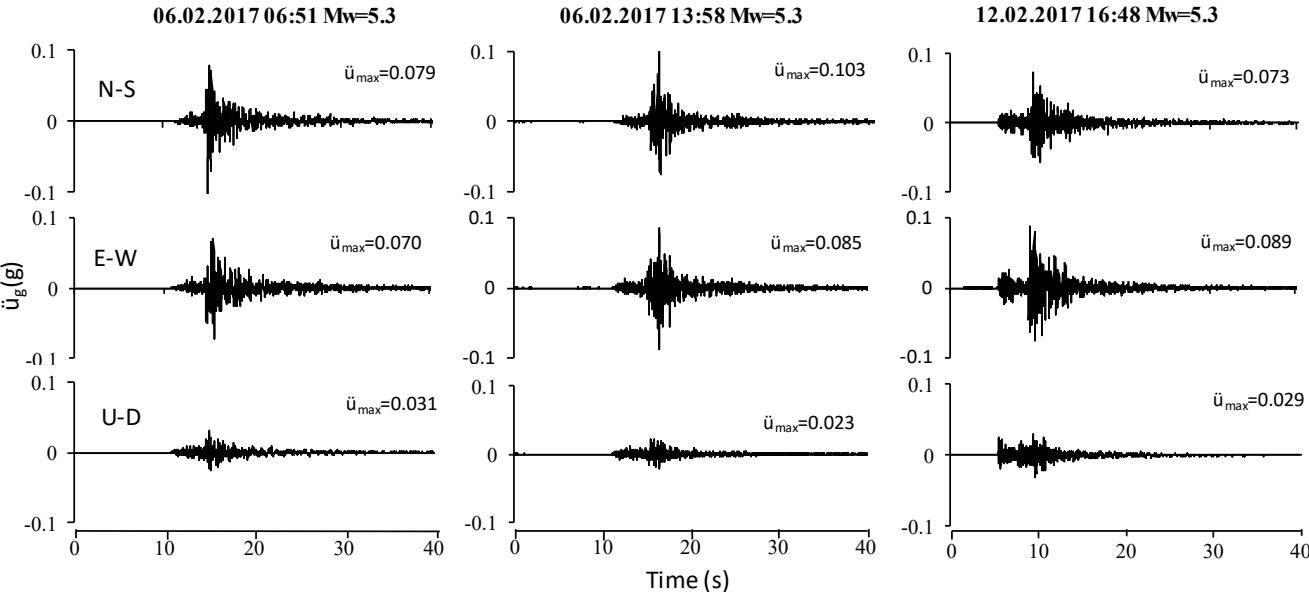

**Figure 6: Three components of ground acceleration (Mw> 5.2) of February 6-24, 2017 Çanakkale Earthquakes**

Response spectra with a damping ratio of 2% and 5% for horizontal components are computed and given in Fig. 7. This figure shows that the earthquake shaking would be most effective on structures having a natural period of approximately up to 0.4 s. The strong ground motion records, taken from Forestry Operation Directorate enabled us to determine the attenuation of the ground accelerations. The peak ground acceleration from the five earthquakes was approximately 0.105 g at the station, which is 24 km away from the epicentre. Similarly, the peak ground acceleration were 0.03 g, 0.009 g, and 0.004 g at Ezine, Bozcaada, and Bayramic stations, which are 31, 33, and 48 km away from the epicentre, respectively

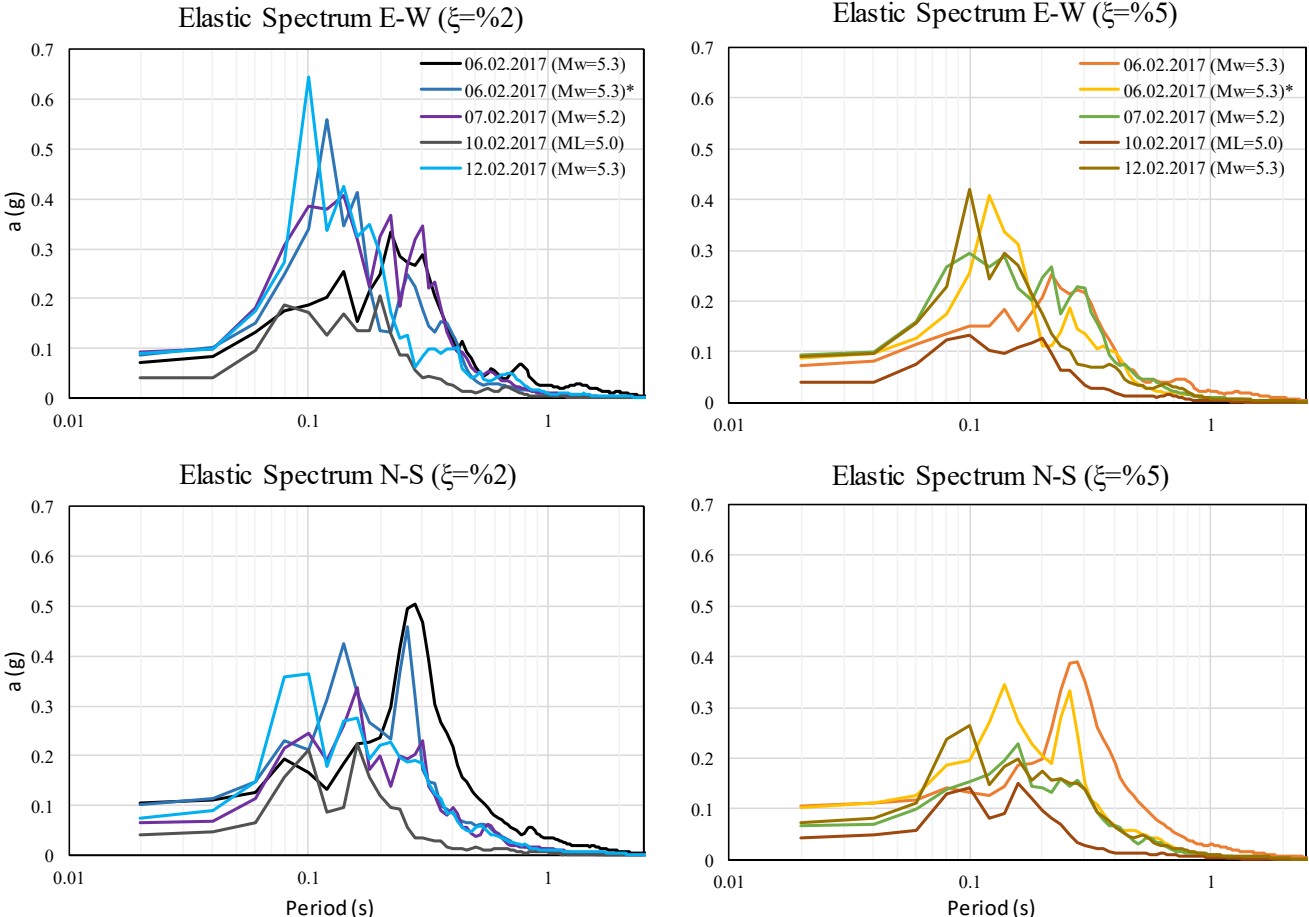

*This earthquake is the second earthquake that occurred in the same day having a magnitude of 5.3

**Figure 7: Elastic acceleration response spectrums for N–S and E–W components of (M$_w$> 5) of February 6-24, 2017 Çanakkale Earthquakes**

5    The peak ground acceleration (PGA) values of Ayvacık records are indicated on the attenuation curve prepared by Gülkan and Kalkan (2002) for M = 5.5 as shown in Fig. 8. The correlation of the observed data with the proposed empirical expression is very satisfactory. It should be noted that because the observed towns are approximately within 3-5 km distance to the epicentre of the earthquakes, the attenuation relation point out that the damaged and collapsed buildings might have experienced 0.2 g and 0.25 g PGA during the earthquakes for rock and soil site conditions, respectively. When elastic response spectra calculated

10   by the Earthquakes and attenuation is considered, the results show that the maximum acceleration exciting the buildings might reach a maximum of 0.25g in the reconnaissance area. On the other hand, the damping ratio can reach a maximum of 5% for such masonry and adobe structures according to the Turkish Earthquake Code, however, design acceleration is offered as 0.5g in this region for masonry buildings. This comparison is the best evidence we have indicating that damaged or collapsed buildings did not receive any engineering service or were not built according to code in force at the time they were built

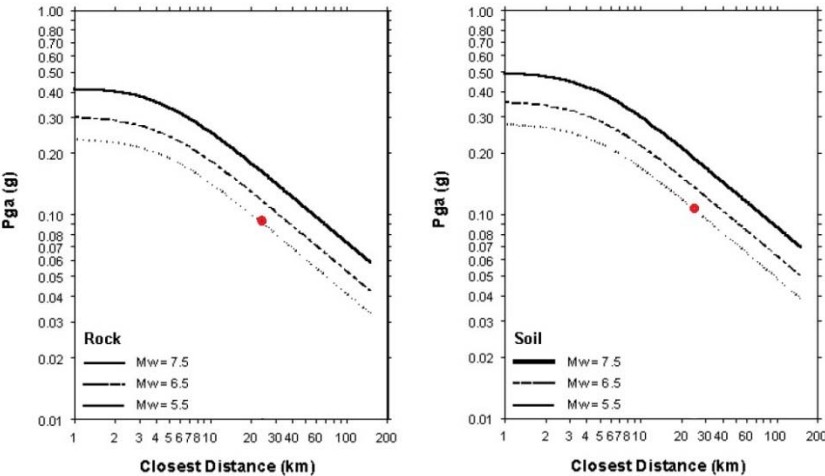

**Figure 8: Curves of peak acceleration versus distance for magnitude 5.5, 6.5 and 7.5 earthquakes at rock and soft soil sites (Gülkan and Kalkan, 2002)**

## 4 Evaluation of Damages

The damages were investigated in two separate subsections. In the first subsection, damage distribution according to affected villages and structures was addressed; while in the latter, occurred damage modes and their technical reasons were evaluated.

### 4.1 Damage distribution

Since the energy release was relatively very small compared to the earthquakes that occurred on NAFZ or on EAFZ, the other most active zone of Turkey, no RC structures collapsed in the area other than the poorly constructed stone masonry dwellings in rural areas. According to the data obtained from Çanakkale Provincial Directorates of Environment and Urbanization, in twenty-nine villages, there were about 2705 damaged or collapsed buildings out of 5790 structures while 3083 structures did not suffer any damage. According to official estimates, within the affected area, a total of 1470 (25%) structures (including apartments, houses, barns, offices, stores and haylofts) were heavily damaged or collapsed, and 1235 (22%) structures suffered medium or minor repairable damage. Moreover, a total of 3083 (53%) structures did not suffer any damage. The locations of twenty-nine villages together with the epicentres of the studied earthquakes, their magnitudes, and PGAs are given in Figure 9, while the number of damaged structures and damage ratios within these villages are given in Figures 10 and 11, respectively. It can be seen from Figure 9 that Taşağıl, Yukarıköy and Çamköy, as well as Gülpınar are close to the epicentres of earthquakes, although structures located in the town of Gülpınar experienced significantly less damage than other villages close to the epicentres (Figure 11). These results may relate to the construction techniques and development level of Gülpınar, which are more improved than the other villages.

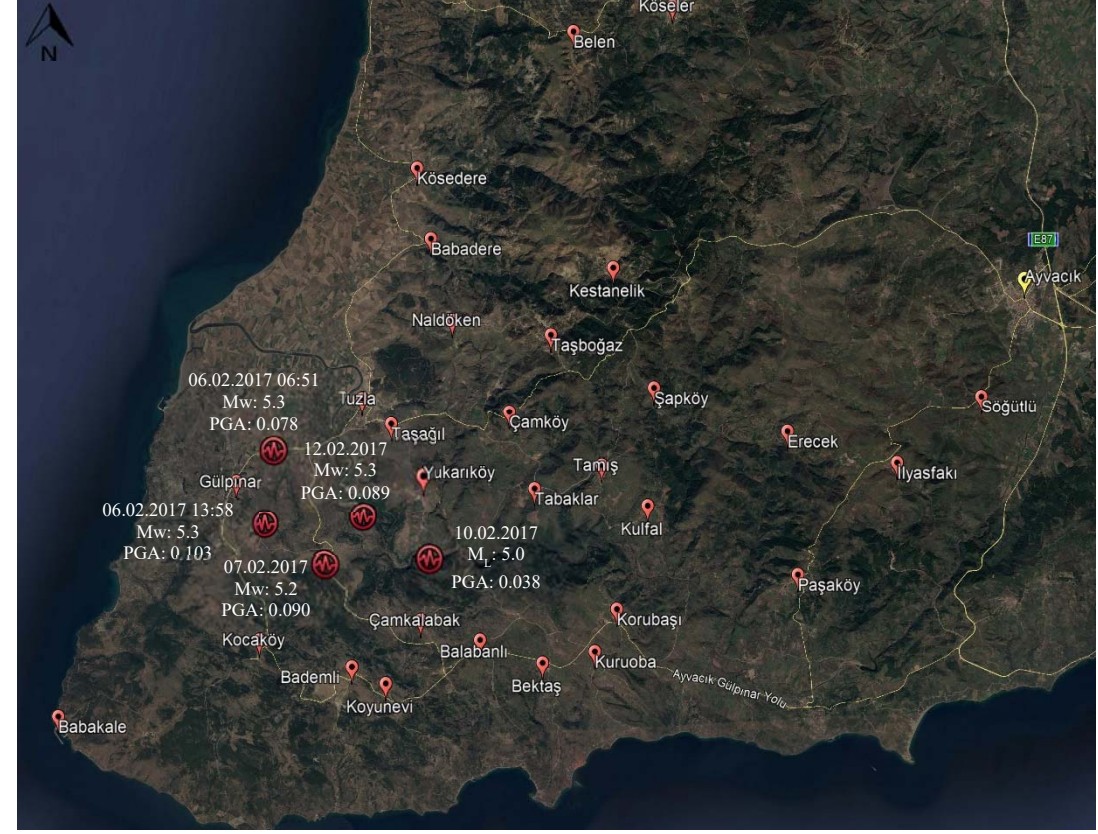

**Figure 9: Villages affected by Ayvacık earthquake swarm and locations of investigated earthquakes**

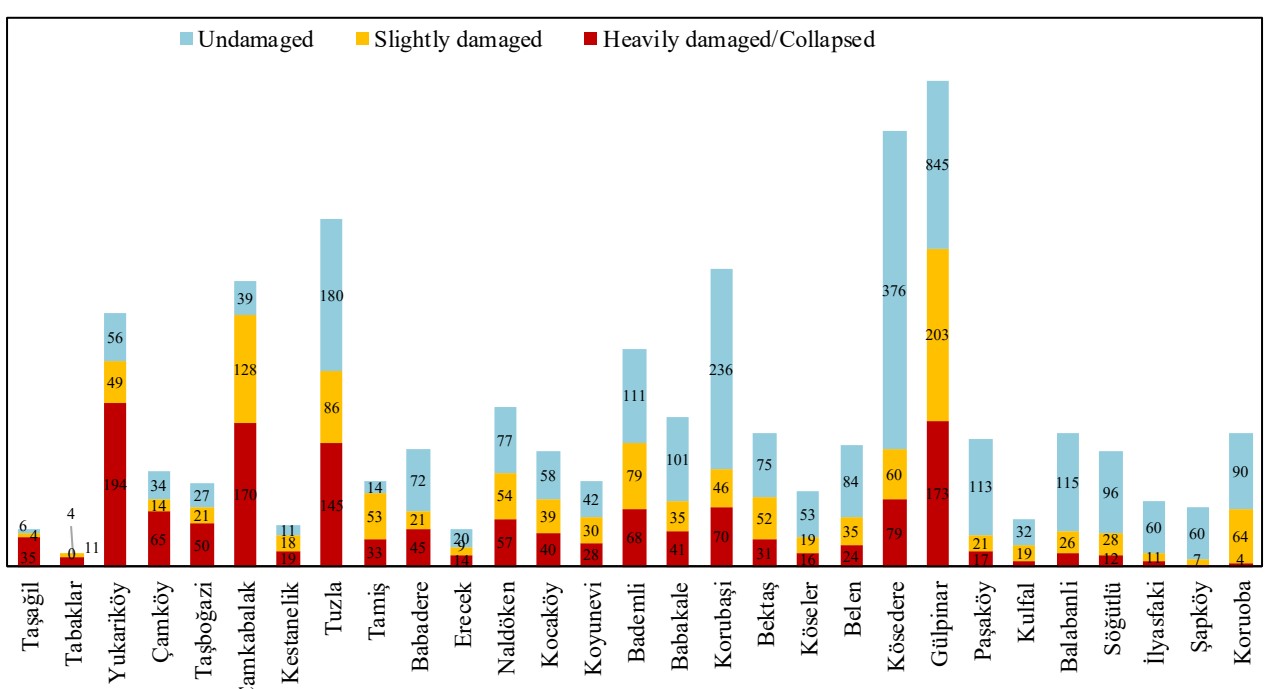

**Figure 10: Number of buildings according to damage level due to Ayvacık earthquake swarm**

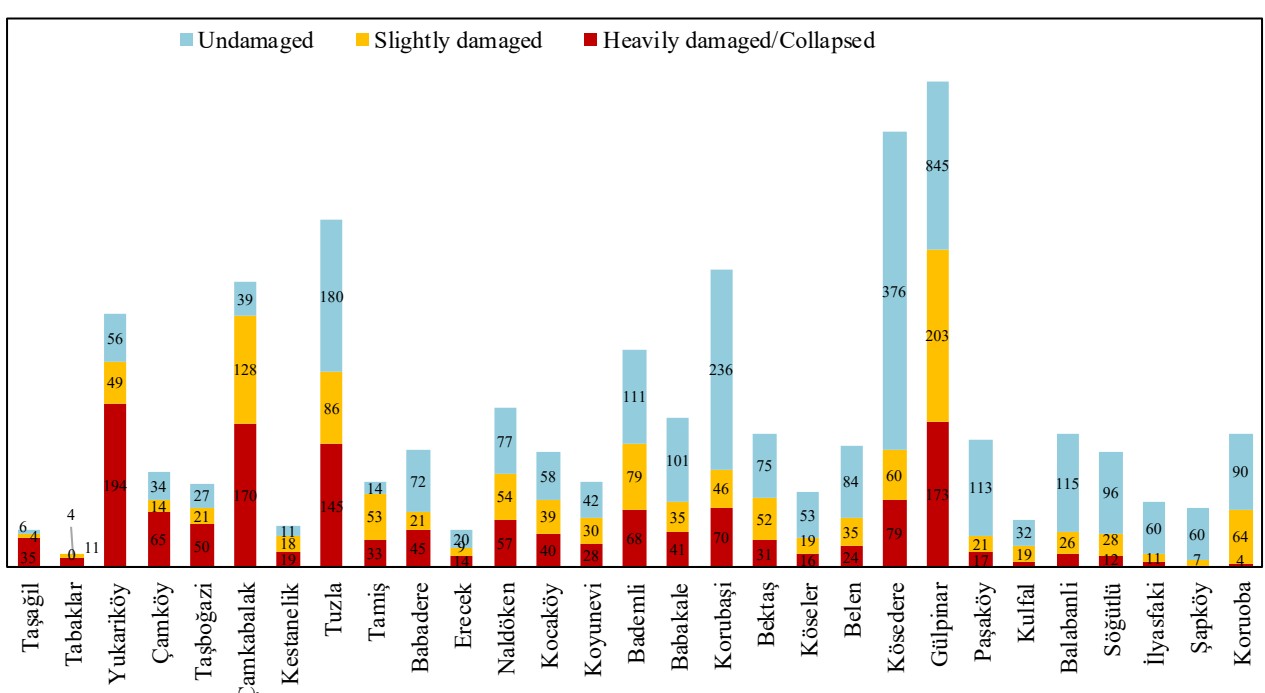

**Figure 11: Damage ratios in Villages according to damage level due to Ayvacık earthquake swarm**

Distribution maps of buildings in percentages according to damage levels are given in Figures 12, 13 and 14, respectively. These figures clearly indicate that the percentage of heavily damaged/collapsed structures in Gülpınar was lower than other villages close to the epicentres that suffered significant damages. The reason for this can be explained by Gülpınar being a historical town centre in the region, therefore the town contain cultural heritage sites. The differences in terms of cultural accumulation and development level between Gülpınar and other villages subsequently affect the quality of construction. Thus, structural damage was more prominent in the villages with relatively low economical development, and where there are no engineered buildings as observed by the authors.

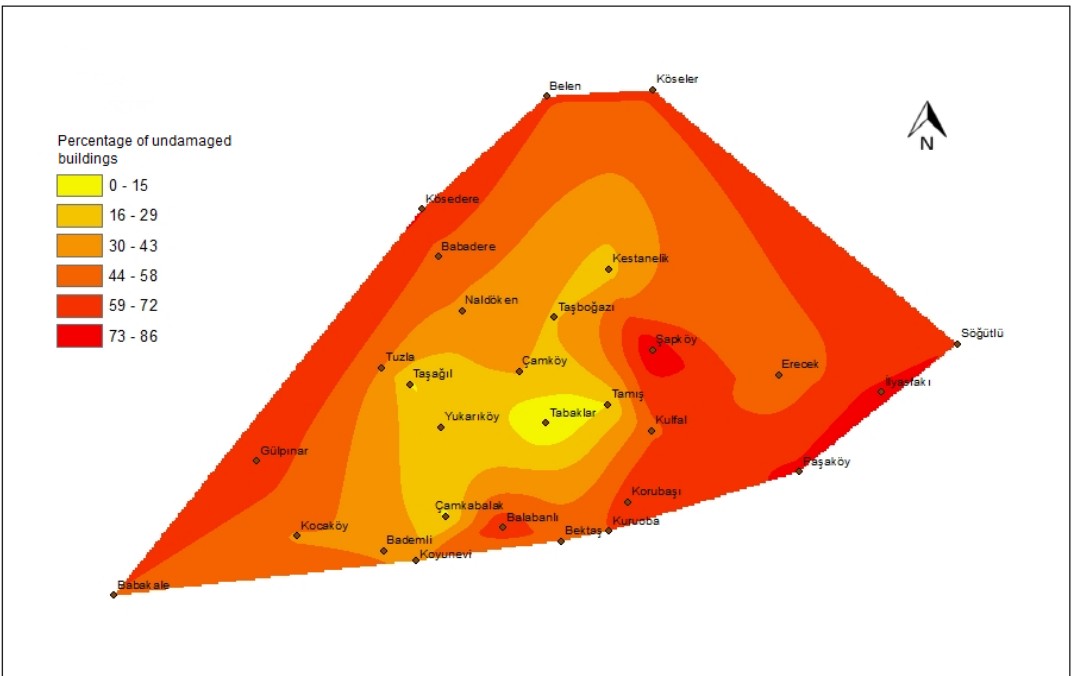

**Figure 12: Distribution maps of undamaged buildings in percentage**

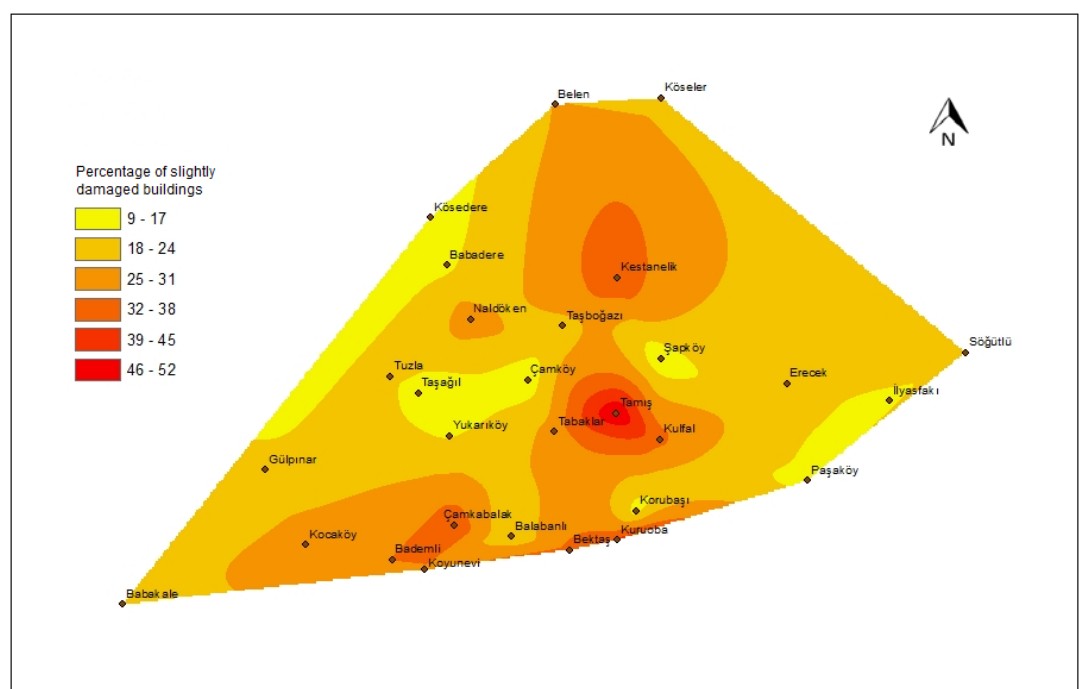

**Figure 13: Distribution maps of slightly damaged buildings in percentage**

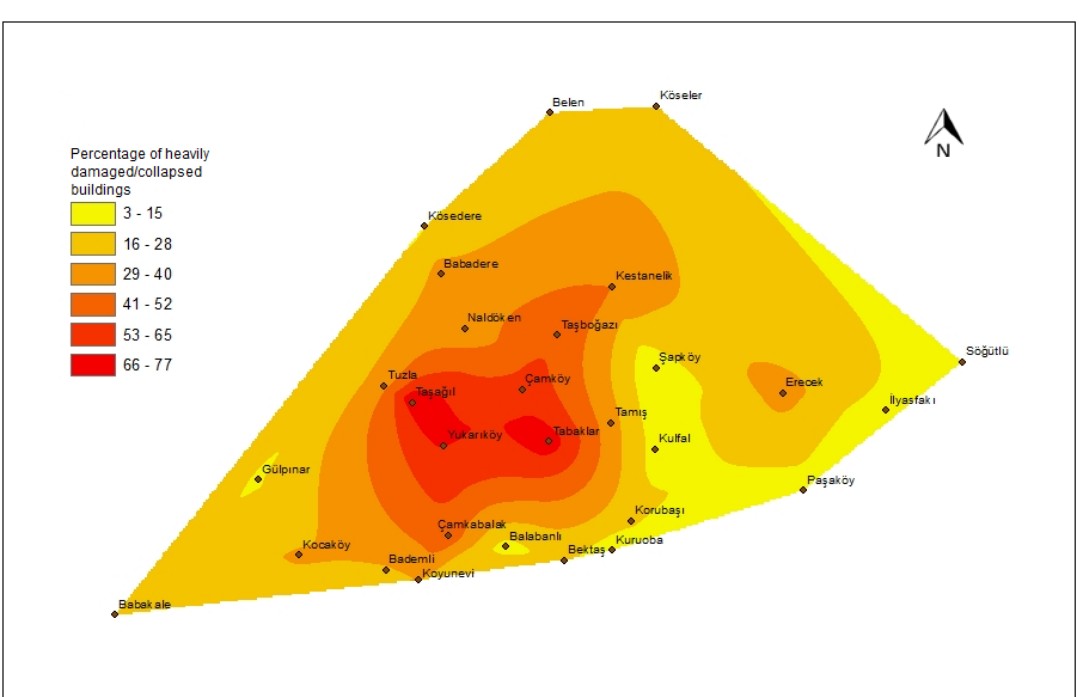

**Figure 14: Distribution maps of heavily damaged/collapsed buildings in percentage**

Distribution maps mentioned above are created for damage ratios according to damage levels of all structures. However, evaluation of damage levels according to structure may introduce a new perspective in interpreting the damages. Besides, such a parametric study may be a guide in order not to repeat similar mistakes when reconstructing structures with a high heavily damaged/collapsed ratio according to structure. Damage ratios according to six structure are generated in Figure 15 with the support of Çanakkale Provincial Directorates of Environment and Urbanization. As can be seen from the figure, the construction practices applied on Haylofts and Barns should be substantially revised in order to minimize damages from a potential similar earthquake. Because most of haylofts and barns are constructed using stone without mortar and also net span between walls of the structures were high. On the other hand, the techniques used on structures having a heavily damaged ratio of approximately 25%, such as stores, houses and apartments may be reviewed and enhanced according to technical deficiencies mentioned in the next section. It can be seen that office structures experienced relatively less damage compared to other structure. Thus, it can be said that construction of office structures was performed more in line with the conditions required by TEC' 2007.

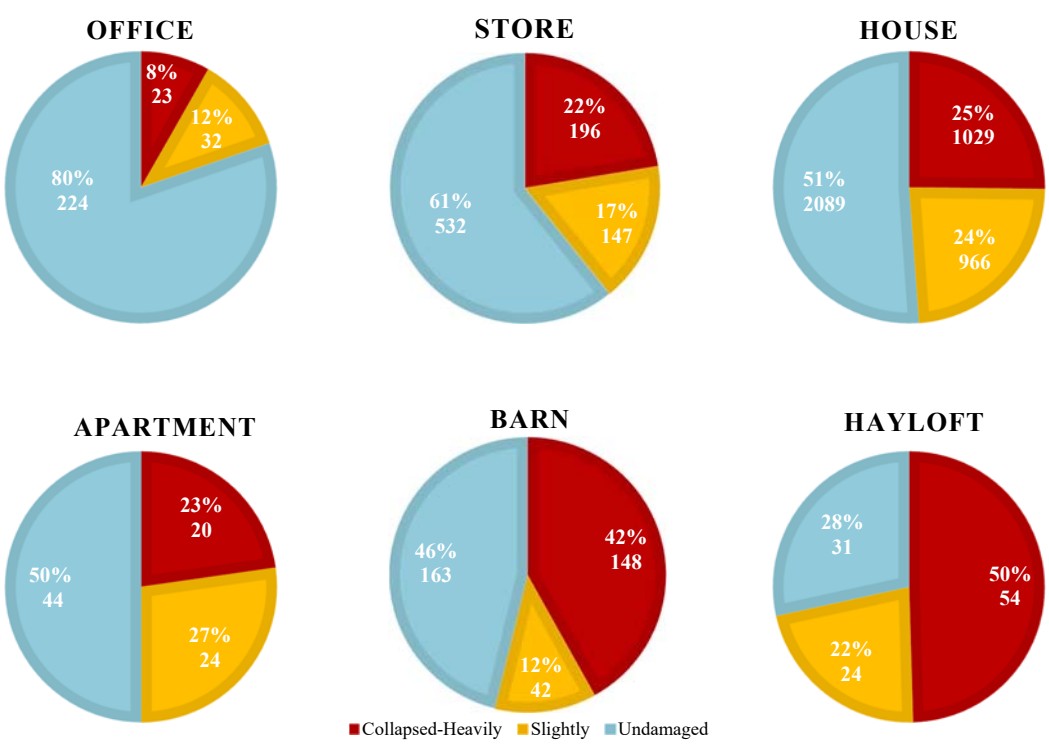

**Figure 15: Damage ratios and levels according to structure**

## 4.2 Damage profile

Failure mechanisms observed during the 2017 Çanakkale earthquake swarm were also observed in other recent moderate earthquakes in Bala (ML=5.5), Doğubeyazıt (ML=5.1), Dinar (ML=5.9) and so on (Tezcan, 1996; Bayraktar et al., 2007; Adanur, 2008; Ural et al., 2012). Adanur (2010) showed that based on the investigations after 20 and 27 December 2007 Bala (Ankara) earthquakes, masonry buildings were built in three types in the affected area: (1) stone masonry buildings with walls made of natural shaped stones, (2) stone masonry buildings with walls made of cut stones, and (3) mixed masonry buildings with walls made of masonry materials like stones and mud bricks, stones and bricks, or stones and briquette. From all, a total of 945 buildings were heavily damaged or collapsed in Bala. Bayraktar et al. (2007) reported that 1000 buildings were affected from the earthquake in Doğubeyazıt. Similar to the above-mentioned studies, thus far, experiences from such moderate earthquakes in rural areas of Turkey have shown that even low-moderate earthquakes may cause significant damages on non-reinforced masonry structures (Fig. 16). This type of construction is among the most vulnerable type of buildings during an earthquake. Even under moderate lateral forces, such a masonry structure is damaged or collapsed due to lack of shear strength, improper interlocking mechanism and/or poor bonding between stone-stone or stone-mortar. In this case, shear failure is unavoidable in the planes forming diagonal cracks along the wall due to workmanship defects. Furthermore, when the wall is not designed with any engineering rule in mind, catastrophic and rapid collapse occurs in out-of-plane bending mode.

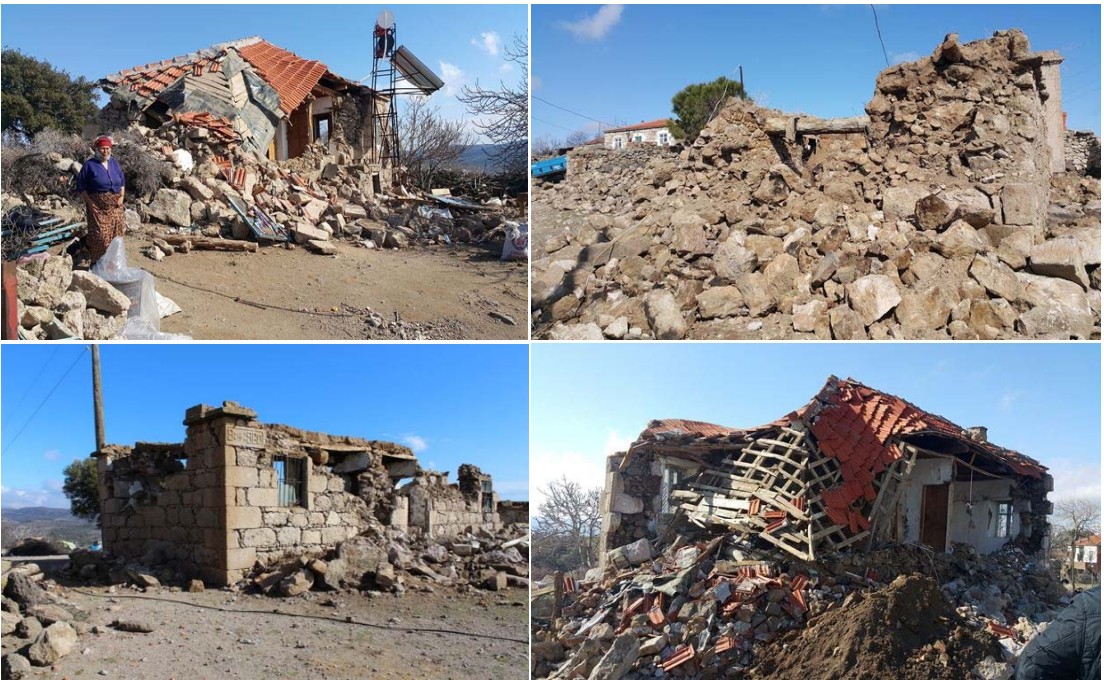

**Figure 16: Examples of totally collapsed structures from Ayvacık, Çanakkale 2017 earthquake swarm**

In addition to the general failure mode mentioned above, the technical causes of damage and crashes observed in the reconnaissance area can be summarized as follows.

- Inadequate interlocking among the stones

In the rural areas of Turkey, the construction of dwellings is done by the owner–dweller with the help of craftsmen who live in the area but are not full-time builders. These builders often learn their trade via apprenticeship. Hence, they have their own tools and do not follow any scientific rules on the site, as a result, an outdated or faulty construction technique can stay alive in a small region, and construction becomes highly similar between the dwellings. For example, during field observations, it was understood that even thick mortar or mud was not used as binding agent between stone or masonry units in almost all

damaged houses. Figure 17 is a striking example of heavy damage during earthquakes due to lack of mortar among stones. After a few mild earthquakes this masonry dwelling became unstable.

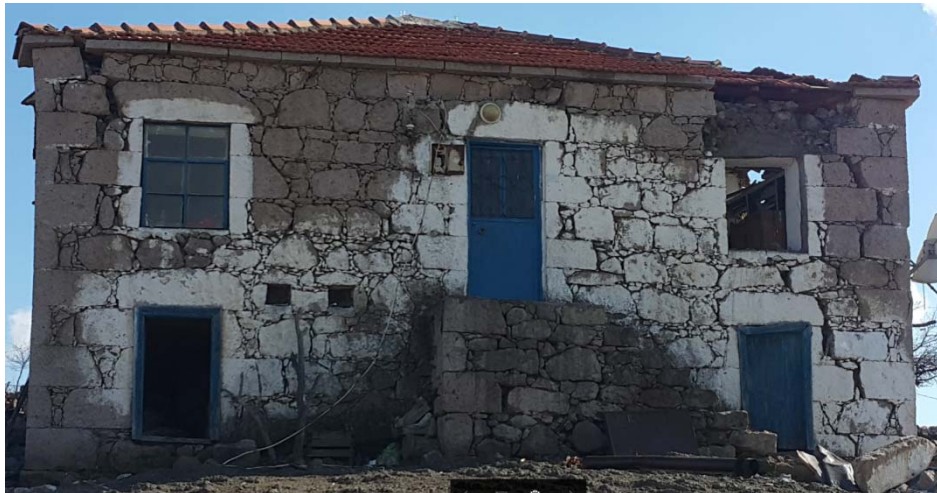

**Figure 17: An example of a damaged dwelling due to inadequate interlocking**

Another damage type observed in the region is outward bulking of walls, which is caused by interlocking deficiency. The reason of this deficiency is the vertical gap between stones creating wall thickness as shown in Fig. 18. In order to prevent this damage, horizontal elements such as 'hatıl' or key stone, which provide integrity to masonry walls, can be vertically used in specific intervals. The key stones or 'hatıl's can provide limited resistance to lateral seismic loads, and thus probably prevent the out-of-plan failure on some part of masonry walls.

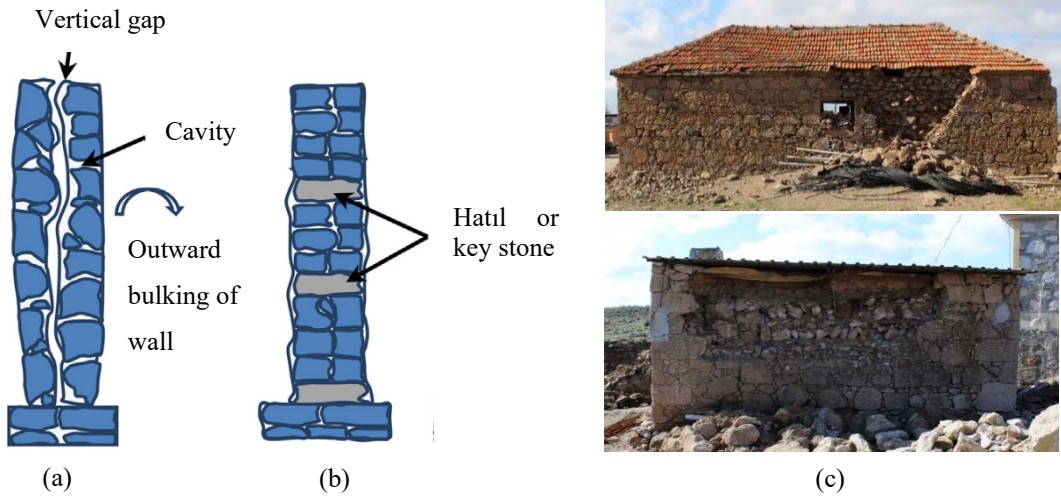

**Figure 18: Schematics of (a) conventional wall section without through stone, (b) wall section with key stones (Sharma, 2016), (c) damages observed in the region.**

Another interlocking damage type is observed at the intersection of perpendicular walls (Fig. 19). One of the walls acts out of plane while the other remains very stiff in plane, resulting in inevitable cracks. This type of damage can either result in gaps developing between the in-plane and out-of-plane wall or vertical cracks may occur in the out-of-plane wall (Tolles et al. 1996). Further stages of this damage may result in out-of-plane failure of gable-end wall. To avoid intersection damage, the interlocking in the corners between perpendicular walls should be properly designed against lateral earthquake forces.

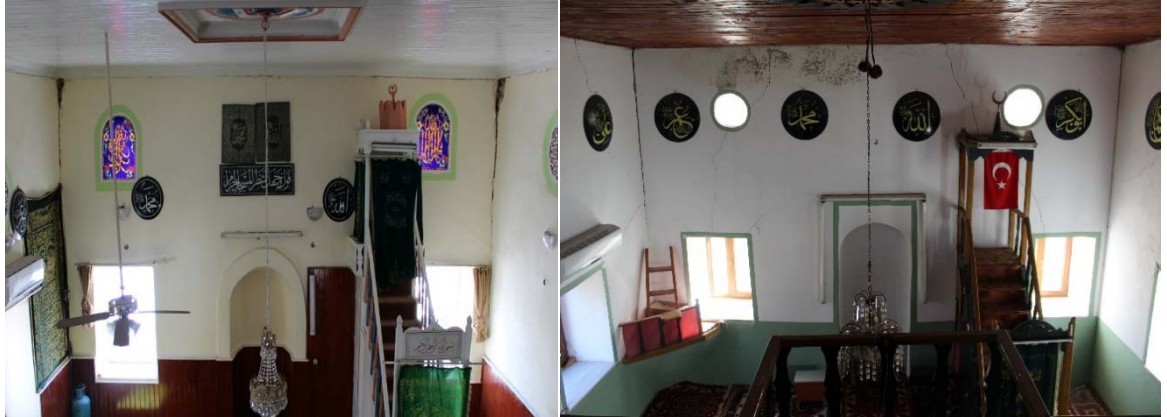

**Figure 19: Observed damages at intersection of perpendicular walls**

- Irregularly designed wall with cavity

The design process of masonry buildings needs more regularity compared to other supporting systems, because resisting system must have continuity and regularity in order to take the stress of the shear forces due to the earthquake. In rural areas,

however, traditional fireplaces have been used in buildings, and they are built within the wall by decreasing the wall thickness or curving the wall outward. In such a case, irretrievable damage is inflicted on the wall because of decreasing shear resistance (Fig. 20). This damage type was observed in different masonry structures in the site. Different cases such as cut stone masonry, stone with plaster, and stone without mortar can be seen in Fig. 20. The common damage type is most likely caused by the lack of craftsmanship or traditional habits.

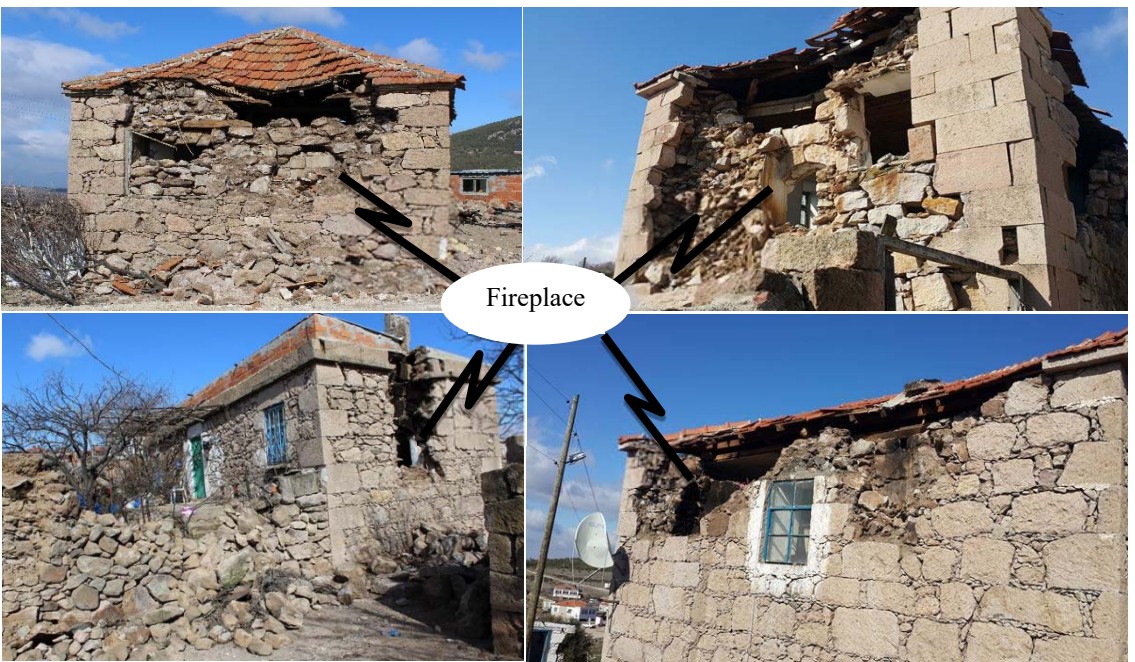

**Figure 20: Examples of out of plane collapse due to wall cavity**

- Heavy Earth Roof

Another important cause of damage is the roofs made from a thick and heavy layer of mud spread upon wooden logs (Fig. 21). This technique is widely used in certain parts of Anatolia where timber is increasingly scarce. These heavy earth roofs are generally hardened by spreading soil with a cylindrical stone. The roof must be thickened more and more over the years to make it more durable against water leakage. Consequently, heavier roofs exert larger shear forces during an earthquake. In the investigated area, the roofs were either supported by beams and indirectly by walls or beams of the inner structure, and the columns were round or sub-round in cross-section, and the trunks were without the barks. This made it virtually impossible to obtain good connections and bearing surfaces between the beams. Such beams were prone to rolling off during motions induced by earthquakes. Moreover, the round ends of the beams exerted loads (to an excessive degree) on the supporting walls beneath them, and resulted in the collapse of the earth roof or walls.

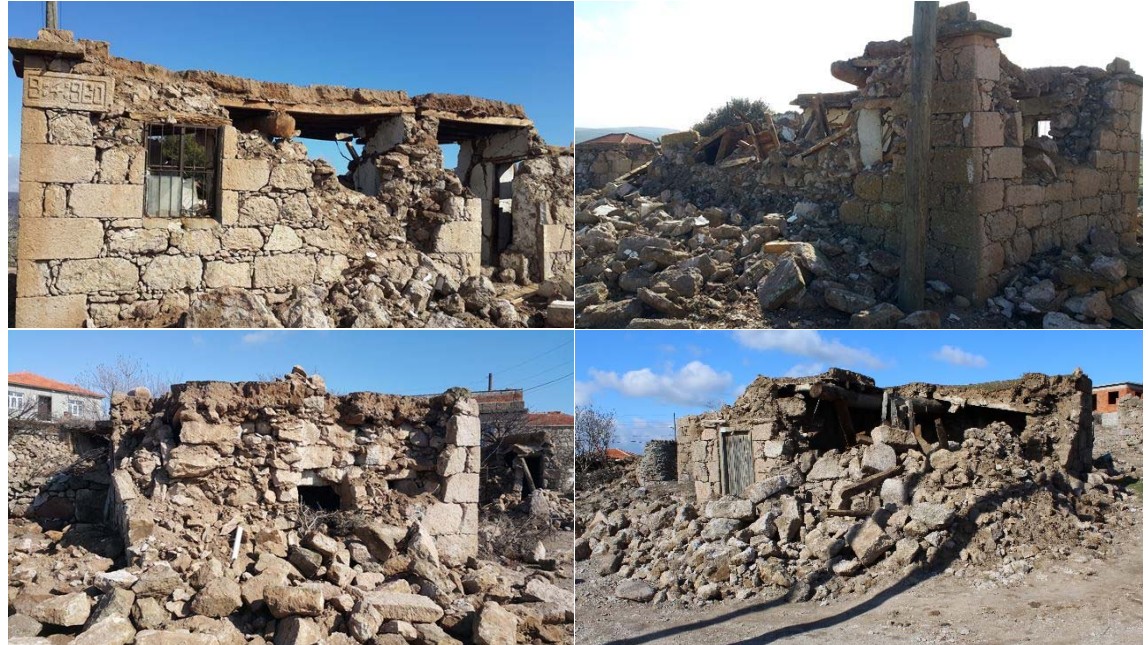

**Figure 21: Examples of collapse due to heavy earth roof**

- Insufficient wall rigidities

In many cases, distinctive diagonal or inclined cracks were observed in load-bearing window piers or walls with low width-to-height ratios as a result of inadequate shear resistance (Tomazevic, 1999). While bending and shear forces created by a moderate earthquake can be easily resisted by reinforced-masonry with lateral and horizontal elements such as RC or timber (Fig. 22), the dwellings made from stone with no mortar cannot resist these forces. This construction defect causes in-plane failures by means of excessive shear force or bending or out of plane failure by bending depending on the aspect ratio of the unreinforced masonry elements.

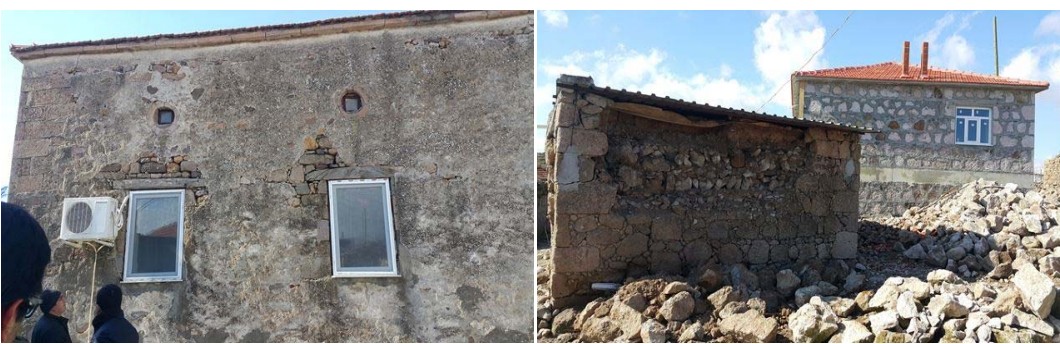

**Figure 22: Examples of undamaged dwellings**

Many weak masonry walls without mortar had diagonal or inclined shear cracking as a result of cyclic shear forces exerted during the earthquakes (Fig. 23). But this diagonal shear cracking does not necessarily lead to total collapse. However, collapse may be inevitable if the triangular wall blocks on each side of a full diagonal crack become unstable by substantially losing their interlock or friction resistance along the cracks (Fig. 24). Similar failures have previously been reported around the world (Ural et al., 2012; Klingner, 2006).

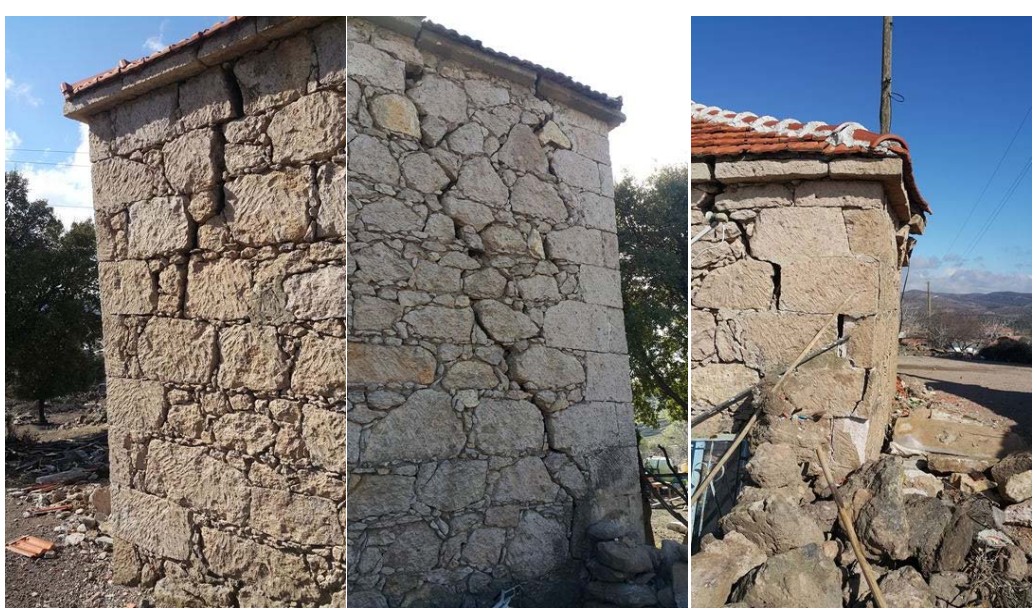

**Figure 23: Examples of diagonal shear cracking**

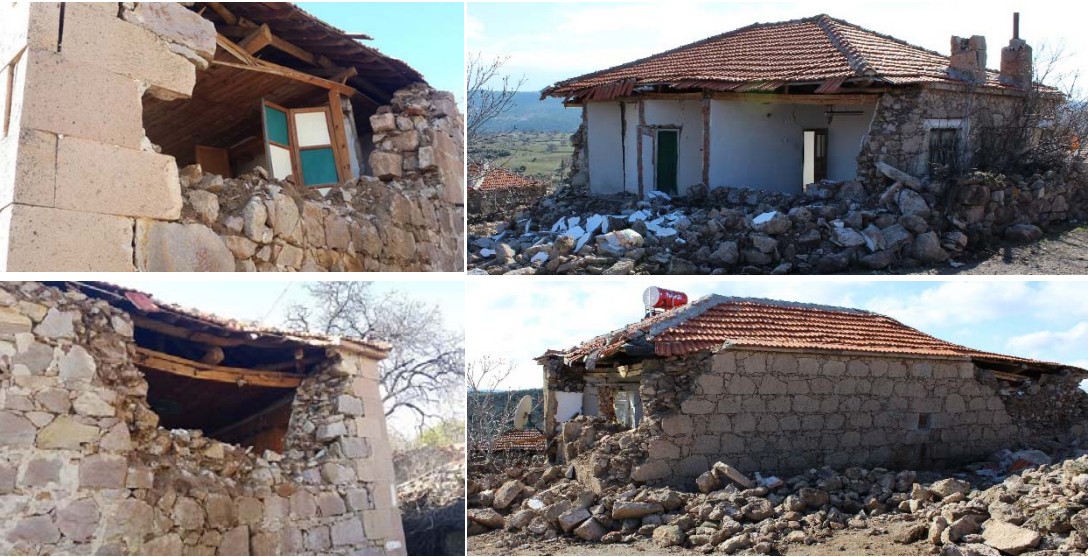

**Figure 24: Out of plane failures due to improper wall thickness and/or height-length ratio**

There were no industrial buildings within Ayvacık, and no damage was observed along the highway or at bridges. There were no reported landslides, or rock fall.

## 5 Conclusions and Recommendations

The aim of this paper is (1) to evaluate the characteristics of earthquakes, (2) to scrutinize the damage distribution in terms of villages and structure types and (3) to investigate the damage and collapse mechanisms observed in buildings during a rarely occurred event called earthquake swarm that struck Ayvacık, Turkey, between 06 and 24 February 2017. This earthquake swarm contained almost 2000 earthquakes with some moderate earthquakes (Mw > 5.0). The properties of these earthquakes with respect to civil engineering such as peak ground acceleration, response spectrum were specified. Although the determined elastic spectrum remained under the design spectrum of TEC (2007), significant damages and failures in many masonry structures were observed in the reconnaissance area. The reasons of these damages and failures observed in the survey can be explained as follows: (1) close proximity of damaged buildings to the epicentre of earthquakes, (2) influence of pre-existing cracks on the performance of buildings due to many earthquakes occurring in a short period of time, (3) deficiency of construction process including poor workmanship and material quality, construction without any scientific rule or code, and lack of bonding or connection between structural elements. On the other hand, damage distribution/ratio decreased as the distance from the epicentres of earthquakes increased, except for Gülpınar.

In conclusion, the authors have some opinions/recommendations about damaged structures in the effected region and structures in other rural regions located in seismic hazard areas. (a) it is recommended by the authors that the construction practice (such as using stone without mortar) commonly used in the affected region causing damage and resulting in failure of buildings, should be avoided. In addition, new structures in the region must be constructed according to TEC' 2007. (b) It is rather hard to find an available retrofitting technique for such a heavy structure that has no connections between its elements. Even if one or more retrofitting techniques could be applied to the structure, the cost of retrofitting such a structure will be possibly higher than constructing a new structure. Thus, retrofitting these damaged structures may not be logical and economic according to authors' observations. (c) The findings in this study indicate that urban transformation started in the cities of Turkey due to seismic risk is also necessary in rural regions, especially in high seismic zones.

## Acknowledgement

We thank Çanakkale Provincial Directorates of Environment and Urbanization for sharing the damage data of the villages, and also appreciate the sincere contribution of Associate Professor Uğur AVDAN from Anadolu University in preparing the distribution maps of damage ratios.

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
