# Peer review of "Damages during February, 6-24 2017 Çanakkale earthquake swarm"

_Natural Hazards and Earth System Sciences, 2017_

## Referee Comment (RC1) · S. Kundak (Referee) · 1 Aug 2017

The manuscript gives information on field survey aftermath EQ swarm, so that it can be evaluated as a preliminary report rather than a research article. I would have some suggestions to authors in developing their manuscript.

1. General composition: The given theoretical background is quite fragmented. An acknowledgement on EQ damage in traditional houses should be given referring past events in both national and international scene. In more detail, a general review of the seismicity of survey region (including consequences of past events) is expected to be given.

2. Presentation of survey area: In the text, authors mentioned about affected villages

where I assume that the research has been conducted in. However, it is not clear how many villages have been studied, how many buildings are in those villages and what is the damage ratio in each. It would be better that the authors produce survey area map(s) indicating epicenter, location and damage ratio in those villages. Furthermore a schematic map might be produced to show PGA distribution. In the text, economic status of residents is mentioned as one of the root causes for damage level (page 7, line 16). If this is a relevant determinant, the authors should give a detailed information on socio-economic status of the survey area.

3. Details: The authors should keep in mind that this manuscript addresses to international readers who are not likely familiar with Turkish abbreviations. For instance: AFAD is given in international papers as DEMP (Disaster and Emergency Management Presidency); MTA is MRE (General Directorate of Mineral Research and Exploration). The authors mentioned Turkish Earthquake Code (TEC). The abbreviate name of this reference is Turkish Earthquake Resistant Code and full name is Specification for Building to be built in Seismic Zones (Not disaster zones. According to the Turkish Regulations, once a region is declared as "disaster zone" no building development is permitted, so that, logically, it is not possible to have a building codes for new development). In the page 5, line 6, it sounds that the survey area zonning is released in 2007's document. However, the Earthquake Zonning Map of Turkey was produced in 1996. In the page 7, line 23, the year of the reference is not given.

4. Conclusion & recommendation: It is very well known that old and low quality (both in material and engineering aspects) buildings are vulnerable to seismic tremors. The recommendations should be beyond re-phrasing "avoid those buildings". Most of the villages are dated back 1970s and before, it is not likely to evaluate them according to the newest building codes. Furthermore, there is always doubt how much they had fit to the old building codes. I suggest authors to focus how traditional rural domestic buildings would be retrofitted using local knowledge and local materials. This approach would give a valuable contribution to the research field.

---

## Referee Comment (RC2) · Anonymous Referee #2 · 1 Aug 2017

I read the paper and the first Referee Comment done by S. Kundak, I agree completely with this comment because it describes most of the observations I wanted to write. In addition, I want to underline that the English level of the text is very poor and there are a lot of errors, so a very important revision in that sense is necessary for a further publication. Another observation is about the figures, that are often too little to be clear.

---

## Author Comment (AC1) · 18 Oct 2017

Response to Dr. Kundak; We would like thank to referee for her valuable comments. According to the Dr Kundak's comments authors tried to improve some part of the study and revised the paper. We believe that most part of the referee's comments were attended and with considerable improvements on the written text. Some revisions are presented in the following pages along with explanations. We have tried our best to accommodate all scholarly comments provided by the reviewer and clarify the ambiguities. We also scrutinized the whole paper to overcome the arisen unclearness through the paper. We hope the revised paper satisfy the standards of your board.

1. General composition: The given theoretical background is quite fragmented. An

acknowledgement on EQ damage in traditional houses should be given referring past events in both national and international scene. In more detail, a general review of the seismicity of survey region (including consequences of past events) is expected to be given.

Author Reply:

- The introduction of manuscript is rearranged and divided into two sections and new subtitle "Seismicity of the Region" was added to the text.

- EQ damage in traditional buildings is given in damage profile (Section 4) with examples of current earthquake damages (Page 7). Damage types, explained widely in Section 4, already include some damage types of past earthquakes. Thus, authors did not prefer to give them in Introduction.

- As referee commented, the seismicity of the region is added to the manuscript as below in section 2: "In this study, investigated region, north-west Anatolia, as both land and sea, is one of the most important active seismic and deformation region between Eurasian and African tectonic plates. The region is affected from both strike-slip tectonic regime which is a general characteristic of NAFZ and extensional regime of west Anatolian block. The most effective earthquake in instrumental period (after 1900) around the region are Aegean Sea earthquake (M=7.2) occurred in 1981, Ayvacik-Çanakkale earthquake (M=7.0) in 1919 and Edremit gulf earthquake (M=6.8) in 1944."

2. Presentation of survey area: In the text, authors mentioned about affected villages where I assume that the research has been conducted in. However, it is not clear how many villages have been studied, how many buildings are in those villages and what is the damage ratio in each. It would be better that the authors produce survey area map(s) indicating epicenter, location and damage ratio in those villages. Furthermore, a schematic map might be produced to show PGA distribution. In the text, economic status of residents is mentioned as one of the root causes for damage level (page 7, line 16). If this is a relevant determinant, the authors should give a detailed information

on socio-economic status of the survey area.

Author Reply:

- The epicenters of earthquakes are given in Fig 1. The locations (latitude and longitude) of those are added to Table 1. PGA of earthquakes are also given in Table 1. Furthermore, a new map including locations of villages, epicenters of earthquakes, their magnitudes and PGAs are added to the manuscript as Figure 9.

- Two different Figures (Figure 10 and 11) are added to manuscript showing number of damaged buildings and damage ratios according to damage level (damaged, slightly damaged, undamaged) for all villages.

- Schematic maps of showing damage distribution for three damage levels are included (Figure 12-13-14)

- In page 7, line 16, Authors mentioned the economic level of survey area as: "Thus, the structural damage was concentrated mainly in the towns which have relatively very low economical level and where there are not any engineered buildings observed by the author."

The low economic level of the area is accepted as a general opinion of Authors at the end of in-situ investigations. Authors think that socio-economic status of the survey area is out of the aim of the paper.

3. Details: The authors should keep in mind that this manuscript addresses to international readers who are not likely familiar with Turkish abbreviations. For instance: AFAD is given in international papers as DEMP (Disaster and Emergency Management Presidency); MTA is MRE (General Directorate of Mineral Research and Exploration). The authors mentioned Turkish Earthquake Code (TEC). The abbreviate name of this reference is Turkish Earthquake Resistant Code and full name is Specification for Building to be built in Seismic Zones (Not disaster zones. According to the Turkish Regulations, once a region is declared as "disaster zone" no building development is permitted, so

that, logically, it is not possible to have a building codes for new development). In the page 5, line 6, it sounds that the survey area zonning is released in 2007's document. However, the Earthquake Zonning Map of Turkey was produced in 1996. In the page 7, line 23, the year of the reference is not given.

Author Reply:

- The referee's comments on abbreviations to are corrected in the text and references.

- For the full name of Turkish Earthquake Code, Disaster zone term is revised as Seismic Zones.

- In the page 5, line 6, mentioned sentence is revised as: "According to earthquake zoning map of Turkey, prepared by General Directorate of Disaster Affairs in 1996, the seismic zone of the city of Çanakkale is classified as 1, where the probability of exceeding an effective peak ground acceleration of 0.4g is 10 percent in 50 years or the return period is 475 years (TEC 2007)."

- Referee's offer on changing "Turkish Earthquake Code" to "Turkish Earthquake Resistant Code" may be a choice. But "Turkish Earthquake Code" is also commonly preferred in literature. Some of articles are listed below:

1. Binici, B., Ozcebe, G., & Ozcelik, R. (2007). Analysis and design of FRP composites for seismic retrofit of infill walls in reinforced concrete frames. Composites Part B: Engineering, 38(5), 575-583.

2. Ural, A., DoÄ§angün, A., Sezen, H., & AngÄśn, Z. (2012). Seismic performance of masonry buildings during the 2007 Bala, Turkey earthquakes. Natural hazards, 60(3), 1013-1026.

3. Altunisik, A. C. and Genç, A. F.: Earthquake Response of Heavily Damaged Historical Masonry Mosques after Restoration, Nat. Hazards Earth Syst. Sci. Discuss., https://doi.org/10.5194/nhess-2017-141, 2017 (Accepted paper).

Thus, Authors prefer to use "Turkish Earthquake Code." - The year of the reference is added in the page 7, line 23.

4. Conclusion & recommendation: It is very well known that old and low quality (both in material and engineering aspects) buildings are vulnerable to seismic tremors. The recommendations should be beyond re-phrasing "avoid those buildings". Most of the villages are dated back 1970s and before, it is not likely to evaluate them according to the newest building codes. Furthermore, there is always doubt how much they had fit to the old building codes. I suggest authors to focus how traditional rural domestic buildings would be retrofitted using local knowledge and local materials. This approach would give a valuable contribution to the research field.

Author Reply:

- In this study, there were not any evaluation of damaged or collapsed building according to TEC 2007. Authors wanted to emphasize that although max acceleration of occurred earthquakes (Fig.7) is very lower than Response Spectrum of current code (TEC 2007), several buildings have experienced damages and many of them are collapsed.

- As mentioned in conclusion, most of damaged structures in the affected area are constructed with poor workmanship and material quality, construction without any scientific rule or code and lack of tie or connection between structural elements. Hence, retrofitting these damaged structures may not be logical and economic according to authors observations. Furthermore, one or more retrofitting techniques may require for each damage type mentioned in 4th section of the discussion paper. For this reason, authors consider that retrofitting technique should be investigated extensively rather than suggesting these techniques in conclusion. And this case will cause the extension of the paper and is beyond the scope of the paper.

The last version of the manuscript is added in supplement.

[Figure]

Best regards...

Please also note the supplement to this comment:
https://www.nat-hazards-earth-syst-sci-discuss.net/nhess-2017-245/nhess-2017-245-AC1-supplement.pdf

**Supplement:**

[revised manuscript text omitted]
, 1170 of them suffered medium or minor repairable damage. Also, a total of 1990 structures did not experience any damage. The locations of twenty-nine villages together with epicenters of considered earthquakes, their magnitudes and PGAs are given in Figure 9 while number of damaged structures and damage ratios according to these villages are given in Figure 10 and 11, respectively. Also, distribution maps of buildings according to percentage for damage levels are given in Figures 12, 13 and 14, respectively. According to this figures, single or a few storey non engineered heavy masonry buildings with very poor details, however, along the sloping hills to the west of Gülpınar-Ayvacık, practically survived the earthquake without significant damage (Figure 14). It should be also noted that Gülpınar is relatively close to the epicenter of the earthquake than other town such as Taşağıl, Yukarıköy and Çamköy where dwellings were suffered very high damages. Gülpınar is also historical center in this area and has the cultural heritages, so the differences in terms of the cultural accumulation and development level between Gülpınar and the other villages affect the quality of the construction. Thus, the structural damage

was concentrated mainly in the villages which have relatively very low economical level and where there are not any engineered buildings observed by the authors.

[Figure]

5 **Figure 9 Villages affected from Ayvacık earthquakes swarm and locations of investigated earthquakes**

[Figure]

**Figure 10 Number of buildings according to damage level due to Ayvacık Earthquake swarm**

[Figure]

5 **Figure 11 Damage ratio in Villages according to damage level due to Ayvacık Earthquake swarm**

[Figure]

**Figure 12 Distribution maps of undamaged buildings according to percentage**

[Figure]

**Figure 13 Distribution maps of slightly damaged buildings according to percentage**

[Figure]

**Figure 14 Distribution maps of damaged/collapsed buildings according to percentage**

[revised manuscript text omitted]

---

## Author Response (AR1)

**Response to Dr. Molinari;**

We would like thank to the editor for the time spent in reviewing our article and her valuable comments. Authors tried to present a revised version of the paper, in which most part of the editor's comments were attended and with considerable improvements on the written text. Some revisions are presented in the following pages along with explanations. We have tried our best to accommodate all scholarly comments provided by the reviewer and clarify the ambiguities. We also scrutinized the whole paper and attached new sections to the text to overcome the arisen unclearness through the paper. But It is worth noting that in civil engineering point of view, authors built the goal of the paper to help the engineers to show damages due to local construction types in moderate seismic events and to better understand damages observed in the site. Anyone can easily find similar articles in the literature which give similar results for earthquakes but every observation includes unique damages belonging the area due to the earthquakes and/or special features of the region. In this context we hope the revised paper satisfy the standards of your board.

1. *The first one is the survey. I think that the way in which you conducted the research (how many buildings? In which area? How they were chosen? Why? How many persons involved? Which tools/instruments you implemented? How long does it take?) as well as its objectives (Which are the parameters, aspects, evidences you collected? Why?) must be fully described, also in order to allow other research to repeat the experience and to compare their results with yours. Maps suggested by the first review can support the discussion.*

**Authors Reply:**

*a) How many buildings?*

Number of buildings according to villages are given Figure 10. Moreover, total number of buildings are added to the text as below with updated numbers.

"According to the data obtained from Çanakkale Provincial Directorates of Environment and Urbanization, in twenty-nine villages alone, there were about 2705 damaged or collapsed buildings out of 5790 structures while 3083 structures did not suffer any damage. According to official estimates, within the affected area, a total of 1470 (25%) structures (including buildings, houses, barns, offices, stores and haylofts) were heavily damaged or collapsed, and 1235 (22%) structures suffered medium or minor repairable damage. Moreover, a total of 3083 (53%) structures did not suffer any damage."

*b) In which area?*

The location of the research area is shown with a red star on the tectonic map of Turkey in Figure 5. In addition, locations of investigated 29 villages with earthquakes are given in detail in Figure 9. The Latitudes and longitudes of the earthquakes are given in Table 1 (indicating the area in which the research is conducted).

The authors think that there are enough information explaining the area of the research. However, if the editor has a better or alternative choice, the authors will try to evaluate and improve the value of the paper.

*c) How they were chosen?*

The buildings in selected villages close to the earthquake epicenter are mostly affected from the earthquake swarm. Since almost 6000 buildings were available in the 29 villages, the authors received support from Çanakkale Provincial Directorates of Environment and Urbanization for the official numbers according to damage level (heavily, collapsed etc.). due to this information reconnaissance team including all authors focused on the damaged area. In civil engineering point of view, it is the most important point for the authors that why damages occurred during a moderate earthquake. Observation on the damages and their reasons were tried to express in the manuscript

*d) How many persons involved? How long does it take?*
These questions are answered by revising the first sentence of the last paragraph of introduction as below:

"A field reconnaissance was carried out by four authors immediately after the earthquakes on February 12-17, for a period of five days and the observations were reported in the present paper."

*e) Which tools/instruments you implemented?*

The aim in reconnaissance researches with viewpoint of civil engineering is to observe damage profile and damage level of structural elements of building as well as their causes rather than non-structural elements. For this reason, simple tools are generally used in this type of researches such as laser meter, meter or plumb. The authors do not find necessary to express these simple tools.

*f) Which are the parameters, aspects, evidences you collected? Why?*

One of the objectives of the reconnaissance paper is to investigate and evaluate the damage patterns of buildings and its causes. This is expressed at the last paragraph of the introduction as:

"The objective of field reconnaissance was to record the causes of the damage patterns observed in the buildings, mainly in the rural areas affected by the earthquake swarm."

Another objective is to evaluate the relationship between response spectra and structural damage. This is stated in the 3rd chapter of the paper.

Additionally, damage distribution of structures (Figure 12-14) in terms of villages and structure types as pie charts (Figure 15) according to damage levels are created and inserted to the text in chapter 4.1.

*g) Also in order to allow other research to repeat the experience and to compare their results with yours. Maps suggested by the first review can support the discussion.*

First of all, the authors try to do their best in order to fulfill the editor's advice to improve the quality of the paper. However, this type of papers generally includes damage types and its levels after earthquake in reconnaissance area as well as earthquake characteristics. There are several similar studies in literature such as Sharma et al. (2016), Adanur (2010) and Xiong et al. (2015). Therefore, the scope of the paper contains similar topics to above studies. Maps suggested by the first referee have been already added and explained in chapter 4.1.

**2.** *The second one concerns results. How can I use collected evidences on damage and its cause? E.g. (1) for calibrating new damage models or validate existing one, in order to improve our risk knowledge of the area? If this is the case, why you did not analyse the relation between hazard and vulnerability? (2) to identify the buildings which are most at risks? How? according to which criteria? (3) to suggest mitigation strategies, e.g. retrofitting, better spatial planning, etc.; what results tell us about this? (4) others?*

**Authors Reply:**

**a)** *(1) for calibrating new damage models or validate existing one, in order to improve our risk knowledge of the area? If this is the case, why you did not analyse the relation between hazard and vulnerability?*

It is not aimed in this study either calibrating new damage model or validate existing one. This is not in the field of the authors expertise. But if anyone want to study in this object we could support them for any aspects.

**b)** *(2) to identify the buildings which are most at risks? How? according to which criteria?*

A comprehensive explanation is conducted and added to the paper as below (Figure 15):

"Distribution maps mentioned above are created for all structures regardless of the structure types. However, evaluation of damage levels according to structure types may introduce a new perspective in interpreting the damages. Besides, such a parametric study may be a guide in order not to repeat similar mistakes when reconstructing structures with a high heavily damaged/collapsed ratio according to structure types. Damage ratios according to six structure types are generated in Figure 15 with the support of Çanakkale Provincial Directorates of Environment and Urbanization. As can be seen from the figure, the construction practices applied on Haylofts and Barns should be substantially revised in order to minimize damages from a potential similar earthquake. On the other hand, the techniques used on structures having a heavily damaged ratio of approximately 25%, such as stores, houses and buildings may be reviewed and enhanced according to technical deficiencies mentioned in the next section. It can be seen that office structures experienced relatively less damage compared to other structure types. Thus, it can be said that construction of office structures were performed more in line with the conditions required by TEC' 2007."

**c)** *(3) to suggest mitigation strategies, e.g. retrofitting, better spatial planning, etc.; what results tell us about this?*

Most of damaged structures in the affected area are constructed with poor workmanship and material quality, construction without any scientific rule or code and lack of tie or connection

between structural elements. Hence, retrofitting these damaged structures can not be logical and economic according to authors observations. Furthermore, one or more retrofitting techniques may require for each damage type mentioned in 4th section of the discussion paper. For this reason, authors consider that retrofitting technique should be investigated extensively rather than suggesting these techniques in conclusion. And this case (adding retrofitting techniques) will cause further extension of the paper and is beyond the scope of the paper.

> *"I think that, at present, results (or better evidences) are simply discussed in a narrative form without any critical analysis of them… which can be the really added value of the research. But, above all, what results tell us which is novel with respect to the state of the art?"*

- As we discussed in the manuscript and responses given here, the structure type used in the site is not common types neither for Turkey nor for the other earthquake zones all over the world. So every observed damages i.e failure examples due to improper interlocking mechanism and/or lack of bonding between stone-stone or stone-mortar give us special failure type for such a regional construction. It is almost not possible to see such connection details in earthquake prone-areas. Besides other a lot of information given in the manuscript only this information will attract readers' attention.

> *"At last, a re-organisation of the contents and a revision by a native speaker are required. The poor English and the fragmented organisation of the manuscript do not help the full comprehension of its main contents."*

- For re-organisation of the paper, introduction is divided into two different sections in earlier revision, as suggested by Dr. Kundak. For a full comprehension of the main contents of the paper, as suggested by the Editor, 4th section of the paper is divided into two separate subsections.

The manuscript is scrutinized and rewritten by native speaker according to the comments.

**Best regards...**

[revised manuscript text omitted]
 1235170 (22%) of themstructures suffered medium or minor repairable damage. AlsoMoreover, a total of 30832990 (53%) structures did not experience suffer any damage. The locations of twenty-nine villages together with the epicentersepicentres of considered the studied earthquakes, their magnitudes, and PGAs are given in Figure 9, while the number of damaged structures and damage ratios according towithin these villages are given in Figures 10 and 11,
20  respectively. It can be seen from Figure 9 that Taşağıl, Yukarıköy and Çamköy, etc. as well as Gülpınar are close to the epicentersepicentres of earthquakes, although structures located in the town of Gülpınar town has experienced significantly less significant damage than other villages close to the epicentersepicentres (Figure 11). This resultsThese results may relate

to the construction techniques and development level of Gülpınar, which are more improved than the other villages.

~~Also, distribution maps of buildings according to percentage for damage levels are given in Figures 12, 13 and 14, respectively. According to this figures, single or a few storey non engineered heavy masonry buildings with very poor details, however, along the sloping hills to the west of Gülpınar-Ayvacık, practically survived the earthquake without significant damage (Figure 14). It should be also noted that Gülpınar is relatively close to the epicenter of the earthquake than other town such as Taşağıl, Yukarıköy and Çamköy where dwellings were suffered very high damages. Gülpınar is also historical center in this area and has the cultural heritages, so the differences in terms of the cultural accumulation and development level between Gülpınar and the other villages affect the quality of the construction. Thus, the structural damage was concentrated mainly in the villages which have relatively very low economical level and where there are not any engineered buildings observed by the authors.~~

[Figure]

**Figure 9: Villages affected  by Ayvacık earthquake swarm and locations of investigated earthquakes**

[Figure]

**Figure 10: Number of buildings according to damage level due to Ayvacık Earthquake swarm**

[Figure]

5   **Figure 11: Damage ratios in Villages according to damage level due to Ayvacık Earthquake swarm**

Distribution maps of buildings in percentages according to damage levels are given in Figures 12, 13 and 14, respectively. These figures, clearly indicate that the percentage of heavily damaged/collapsed structures in Gülpınar was lower than other villages close to the epicentres that suffered significant damages. The reason for this can be explained by Gülpınar being a historical town centre in the region, therefore the town contain cultural heritage sites. The differences in terms of cultural accumulation and development level between Gülpınar and other villages subsequently affect the quality of construction. Thus, structural damage was more prominent in the villages with relatively low economical development, and where there are no engineered buildings as observed by the authors. ~~that structures in Gülpınar experienced rare heavily damaged/collapsed level while several structures in villages close to the epicenter suffer significant damage level. The reason of this situation is that , 
[revised manuscript text omitted]

**Response to Dr. Molinari;**

We would like thank to the editor for the time spent in reviewing our article and her valuable comments. Authors tried to present a revised version of the paper, in which most part of the editor's comments were attended and with considerable improvements on the written text. Some revisions are presented in the following pages along with explanations. We have tried our best to accommodate all scholarly comments provided by the reviewer and clarify the ambiguities. We also scrutinized the whole paper and attached new sections to the text to overcome the arisen unclearness through the paper. But It is worth noting that in civil engineering point of view, authors built the goal of the paper to help the engineers to show damages due to local construction types in moderate seismic events and to better understand damages observed in the site. Anyone can easily find similar articles in the literature which give similar results for earthquakes but every observation includes unique damages belonging the area due to the earthquakes and/or special features of the region. In this context we hope the revised paper satisfy the standards of your board.

1. *The first one is the survey. I think that the way in which you conducted the research (how many buildings? In which area? How they were chosen? Why? How many persons involved? Which tools/instruments you implemented? How long does it take?) as well as its objectives (Which are the parameters, aspects, evidences you collected? Why?) must be fully described, also in order to allow other research to repeat the experience and to compare their results with yours. Maps suggested by the first review can support the discussion.*

**Authors Reply:**

a) *How many buildings?*

Number of buildings according to villages are given Figure 10. Moreover, total number of buildings are added to the text as below with updated numbers.

"According to the data obtained from Çanakkale Provincial Directorates of Environment and Urbanization, in twenty-nine villages alone, there were about 2705 damaged or collapsed buildings out of 5790 structures while 3083 structures did not suffer any damage. According to official estimates, within the affected area, a total of 1470 (25%) structures (including buildings, houses, barns, offices, stores and haylofts) were heavily damaged or collapsed, and 1235 (22%) structures suffered medium or minor repairable damage. Moreover, a total of 3083 (53%) structures did not suffer any damage."

b) *In which area?*

The location of the research area is shown with a red star on the tectonic map of Turkey in Figure 5. In addition, locations of investigated 29 villages with earthquakes are given in detail in Figure 9. The Latitudes and longitudes of the earthquakes are given in Table 1 (indicating the area in which the research is conducted).

The authors think that there are enough information explaining the area of the research. However, if the editor has a better or alternative choice, the authors will try to evaluate and improve the value of the paper.

*c) How they were chosen?*

The buildings in selected villages close to the earthquake epicenter are mostly affected from the earthquake swarm. Since almost 6000 buildings were available in the 29 villages, the authors received support from Çanakkale Provincial Directorates of Environment and Urbanization for the official numbers according to damage level (heavily, collapsed etc.). due to this information reconnaissance team including all authors focused on the damaged area. In civil engineering point of view, it is the most important point for the authors that why damages occurred during a moderate earthquake. Observation on the damages and their reasons were tried to express in the manuscript

*d) How many persons involved? How long does it take?*

These questions are answered by revising the first sentence of the last paragraph of introduction as below:

"A field reconnaissance was carried out by four authors immediately after the earthquakes on February 12-17, for a period of five days and the observations were reported in the present paper."

*e) Which tools/instruments you implemented?*

The aim in reconnaissance researches with viewpoint of civil engineering is to observe damage profile and damage level of structural elements of building as well as their causes rather than non-structural elements. For this reason, simple tools are generally used in this type of researches such as laser meter, meter or plumb. The authors do not find necessary to express these simple tools.

*f) Which are the parameters, aspects, evidences you collected? Why?*

One of the objectives of the reconnaissance paper is to investigate and evaluate the damage patterns of buildings and its causes. This is expressed at the last paragraph of the introduction as:

"The objective of field reconnaissance was to record the causes of the damage patterns observed in the buildings, mainly in the rural areas affected by the earthquake swarm."

Another objective is to evaluate the relationship between response spectra and structural damage. This is stated in the 3$^{rd}$ chapter of the paper.

Additionally, damage distribution of structures (Figure 12-14) in terms of villages and structure types as pie charts (Figure 15) according to damage levels are created and inserted to the text in chapter 4.1.

*g) Also in order to allow other research to repeat the experience and to compare their results with yours. Maps suggested by the first review can support the discussion.*

First of all, the authors try to do their best in order to fulfill the editor's advice to improve the quality of the paper. However, this type of papers generally includes damage types and its levels after earthquake in reconnaissance area as well as earthquake characteristics. There are several similar studies in literature such as Sharma et al. (2016), Adanur (2010) and Xiong et al. (2015). Therefore, the scope of the paper contains similar topics to above studies. Maps suggested by the first referee have been already added and explained in chapter 4.1.

> **2.** *The second one concerns results. How can I use collected evidences on damage and its cause? E.g. (1) for calibrating new damage models or validate existing one, in order to improve our risk knowledge of the area? If this is the case, why you did not analyse the relation between hazard and vulnerability? (2) to identify the buildings which are most at risks? How? according to which criteria? (3) to suggest mitigation strategies, e.g. retrofitting, better spatial planning, etc.; what results tell us about this? (4) others?*

**Authors Reply:**

> **a)** *(1) for calibrating new damage models or validate existing one, in order to improve our risk knowledge of the area? If this is the case, why you did not analyse the relation between hazard and vulnerability?*

It is not aimed in this study either calibrating new damage model or validate existing one. This is not in the field of the authors expertise. But if anyone want to study in this object we could support them for any aspects.

> **b)** *(2) to identify the buildings which are most at risks? How? according to which criteria?*

A comprehensive explanation is conducted and added to the paper as below (Figure 15):

"Distribution maps mentioned above are created for all structures regardless of the structure types. However, evaluation of damage levels according to structure types may introduce a new perspective in interpreting the damages. Besides, such a parametric study may be a guide in order not to repeat similar mistakes when reconstructing structures with a high heavily damaged/collapsed ratio according to structure types. Damage ratios according to six structure types are generated in Figure 15 with the support of Çanakkale Provincial Directorates of Environment and Urbanization. As can be seen from the figure, the construction practices applied on Haylofts and Barns should be substantially revised in order to minimize damages from a potential similar earthquake. On the other hand, the techniques used on structures having a heavily damaged ratio of approximately 25%, such as stores, houses and buildings may be reviewed and enhanced according to technical deficiencies mentioned in the next section. It can be seen that office structures experienced relatively less damage compared to other structure types. Thus, it can be said that construction of office structures were performed more in line with the conditions required by TEC' 2007."

> **c)** *(3) to suggest mitigation strategies, e.g. retrofitting, better spatial planning, etc.; what results tell us about this?*

Most of damaged structures in the affected area are constructed with poor workmanship and material quality, construction without any scientific rule or code and lack of tie or connection

between structural elements. Hence, retrofitting these damaged structures can not be logical and economic according to authors observations. Furthermore, one or more retrofitting techniques may require for each damage type mentioned in 4th section of the discussion paper. For this reason, authors consider that retrofitting technique should be investigated extensively rather than suggesting these techniques in conclusion. And this case (adding retrofitting techniques) will cause further extension of the paper and is beyond the scope of the paper.

> *"I think that, at present, results (or better evidences) are simply discussed in a narrative form without any critical analysis of them… which can be the really added value of the research. But, above all, what results tell us which is novel with respect to the state of the art?"*

- As we discussed in the manuscript and responses given here, the structure type used in the site is not common types neither for Turkey nor for the other earthquake zones all over the world. So every observed damages i.e failure examples due to improper interlocking mechanism and/or lack of bonding between stone-stone or stone-mortar give us special failure type for such a regional construction. It is almost not possible to see such connection details in earthquake prone-areas. Besides other a lot of information given in the manuscript only this information will attract readers' attention.

> *"At last, a re-organisation of the contents and a revision by a native speaker are required. The poor English and the fragmented organisation of the manuscript do not help the full comprehension of its main contents."*

- For re-organisation of the paper, introduction is divided into two different sections in earlier revision, as suggested by Dr. Kundak. For a full comprehension of the main contents of the paper, as suggested by the Editor, 4[th] section of the paper is divided into two separate subsections.

The manuscript is scrutinized and rewritten by native speaker according to the comments.

**Best regards...**

---

## Referee Report (RR1)

[referee-annotated manuscript omitted]

---

## Editor Decision (ED1)

Dear Authors,

I have read with interest your manuscript as well as reviewers' concerns and your response.

Although the effort you put in the survey is valuable and the evidence you got from the survey has big potentialities, I think you did not give value to them in the paper whose conclusions, as suggested by the first reviewer, are limited to "avoid old and low quality buildings". At present, such results are very poor for a research paper.

I am quite sceptical that the changes you proposed in response to reviewers' concerns can improve the paper substantially. For this reason, I strongly recommend a deep review of the manuscript which should be directed to two main aspects of the research:

- The first one is the survey. I think that the way in which you conducted the research (how many buildings? In which area? How they were chosen? Why? How many persons involved? Which tools/instruments you implemented? How long does it take?) as well as its objectives (Which are the parameters, aspects, evidences you collected? Why?) must be fully described, also in order to allow other research to repeat the experience and to compare their results with yours. Maps suggested by the first review can support the discussion.
- The second one concerns results. How can I use collected evidences on damage and its cause? E.g. (1) for calibrating new damage models or validate existing one, in order to improve our risk knowledge of the area? If this is the case, why you did not analyse the relation between hazard and vulnerability? (2) to identify the buildings which are most at risks? How? according to which criteria? (3) to suggest mitigation strategies, e.g. retrofitting, better spatial planning, etc.; what results tell us about this? (4) others?
  I think that, at present, results (or better evicdences) are simply discussed in a narrative form without any critical analysis of them… which can be the really added value of the research. But, above all, what results tell us which is novel with respect to the state of the art?

I also thing, as suggested by the referees, that the paper must be better situated in the existing literature on the topic and that the research question(s) must be clearly identified. At last, a re-organisation of the contents and a revision by a native speaker are required. The poor English and the fragmented organisation of the manuscript do not help the full comprehension of its main contents.

---

## Author Response (AR2)

**AUTHOR RESPONSE TO DR. KUNDAK;**

The authors thank to Dr. Kundak, we believe that her significant contributions and recommendations improved the value of the research.

*EDITOR: "I think that, at present, results (or better evidences) are simply discussed in a narrative form without any critical analysis of them… which can be the really added value of the research. But, above all, what results tell us which is novel with respect to the state of the art?"*

*AUTHORS:• "As we discussed in the manuscript and responses given here, the structure type used in the site is not common types neither for Turkey nor for the other earthquake zones all over the world. So every observed damages i.e failure examples due to improper interlocking mechanism and/or lack of bonding between stone-stone or stone-mortar give us special failure type for such a regional construction. It is almost not possible to see such connection details in earthquake prone-areas. Besides other a lot of information given in the manuscript only this information will attract readers' attention."*

- *Unfortunately I cannot agree with the statement of the authors. Once we talk about the traditional structures, we can easily observe similarities either structural type or materials used. Especially, considering traditional structures in Anatolia which had been affected both large cultural basin (from the Balkan Region through Iran) and climatic conditions. For instance, it is easy to find out analogues of timber used structures in Black Sea and Balkan Region. We can also find masonry or other types' of structure in different zones. For sure, it shouldn't be expected to find the exact structural types of the survey area in another single zone, but each sample in the findings of this survey would enlighten different issues at different regions. Therefore, I recommend the authors to re-evaluate their approach, to reveal the novelties in their research and to reflect the outcomes to improve the state of the art.*

The comment we tried to explain from our perspective caused misleading argue. We did not deny similarities, however, definition of the structural system does not depend only the material. This point should not be ignored by civil engineers. The construction techniques used for building in the observed area completely differ from common usage and anyone cannot encounter such a wider usage in other part of Anatolia for human use. One can see such a sloppy construction for only structures in which people do not live (i.e barn, pen, hayloft etc.).

Historically the general trend observed in the earthquake prone areas structural system had been changed and improved against lateral loads (due to earthquakes). So you can easily find its evidence for almost all structural types like timber, masonry etc. For example, the timber house constructed traditionally in the Black Sea region of Turkey have differences from other regions of Turkey. As seen from the figure 1a given below, the typical timber house in east of the Black Sea region does not have any lateral resisting elements, however, as a consequence of experiences there are a lot of diagonal timber elements usage against lateral loads of building structures in figure 1b in Düzce where there has been high seismic activity. Although same material is used for two typical timber houses, structural systems are so different.

Similarly, the techniques for masonry, traditionally used in Turkey, experienced and improved due to earthquake and so there are distinctive differences region by region. As we discussed in the manuscript, in the rural areas of Turkey, the construction of dwellings is done by the owner–dweller with the help of craftsmen who live in the area but are not full-time builders. These builders often learn their trade via apprenticeship. Hence, they have their own tools and do not follow any scientific rules on the site, as a result, an outdated or faulty construction technique can stay alive in a small region, and construction becomes highly similar between the dwellings. For example, during field

observations, it was understood that even thick mortar or mud was not used as binding agent between stone or masonry units in almost all damaged houses. We accept that there are several reasons which can be explained and argued by sociology economy and many other professions. But from our point of view, although the region exposed to high earthquake hazards and experienced earthquakes within recent history, the construction type widely used in the reconnaissance area has characteristic differences. We precisely claim at this point that the observed poor construction, is not shown in another part of Turkey which has high seismic hazard. Figure 2 shows the differences we tried to express, although same material used in both building with poor construction these two structural system are so different.

[Figure]

Figure 1. (a)Typical timber house from East Black Sea region (b)Typical timber house from Duzce

[Figure]

Figure 2. (a) an example traditional masonry in Turkey (b) an example from the region

- *The second point is on the Figure 15 and related paragraph. Structure type is related with the construction, but the authors refer the occupancy of the units/buildings. So that the terminology is misused here. We can understand the different ways of usage such as office, store (shop would fit better), house (housing unit or single housing unit), barn and hayloft. However, it is not clear what "building" refers to. Here I recommend the authors to cross structure types and occupancy to reach some tangible findings. For instance, if the percentage of x structures is homogenously distributed in each occupancy, here the maintenance related to the occupancy would be primary point in damage ratios. Or if the percentage of x structure is higher in haylofts and barns and the damage ratio is also high, the conclusion would be on the problems of the construction/structure type.*

As stated by the referee, the structure type in the text is turn into "structure" and building is revised as "apartment" in Fig. 15. The authors, in the reconnaissance region, observed that most of haylofts

and barns are constructed, similar to house, by using stone without mortar and also net span between walls of the structures were great. This is added to the paper. However, a specific value of damage ratio could not be given because either authors or government agencies did not carry out such a survey.

- *The third point, as I mentioned in my first review, on the conclusion and recommendation part. This part should give concrete remarks referring the findings of the research. However, this section is still weak. So I would repeat my previous comment here: "It is very well known that old and low quality (both in material and engineering aspects) buildings are vulnerable to seismic tremors. The recommendations should be beyond re-phrasing "avoid those buildings". Most of the villages are dated back 1970s and before, it is not likely to evaluate them according to the newest building codes. Furthermore, there is always doubt how much they had fit to the old building codes. I suggest authors to focus how traditional rural domestic buildings would be retrofitted using local knowledge and local materials. This approach would give a valuable contribution to the research field."*

According to Dr. Kundak's recommendations, Conclusions and Recommendations section of the paper is extended as below:

"In conclusion, the authors have some opinions/recommendations about damaged structures in the effected region and structures in other rural regions located in seismic hazard areas. (a) it is recommended by the authors that the construction practice (such as using stone without mortar) commonly used in the affected region causing damage and resulting in failure of buildings, should be avoided. In addition, new structures in the region must be constructed according to Turkish Earthquake Code. (b) It is rather hard to find an available retrofitting technique for such a heavy structure that has no connections between its elements. Even if one or more retrofitting techniques could be applied to the structure, the cost of retrofitting such a structure may be higher than constructing a new structure. Thus, retrofitting these damaged structures may not be logical and economic according to authors observations. (c) The findings in this study indicates that urban transformation started in the cities of Turkey due to seismic risk is also necessary in rural regions, especially in high seismic zones."

We agree with reviewer that it is not possible to find the buildings which meet the new requirements given in the current codes. On the other hand, the code includes almost all experiences and development in certain issue. So we used the code only for the purpose of explaining a specific point. This point may be attractive for reader. However, design acceleration is offered as 0.5g in this region for masonry buildings. The buildings experienced only 0.1g ±0.02. for this purpose, the expression revised as below in the manuscript (the last sentences of section 3).

"This comparison is the best evidence we have indicating that damaged or collapsed buildings did not receive any engineering service or were not built according to code in force at the time they were built."

**AUTHOR RESPONSE TO REFEREE #2**

The authors thank to referee for his/her valuable contribution. The recommendations made by the referee is added to the text.

[revised manuscript text omitted]